# One-Bit Distributed Mean Estimation with Unknown Variance

**Ritesh Kumar**                                                              *ritesharyan622@gmail.com*
*Department of Electrical Engineering*
*Indian Institute of Technology Hyderabad, India*

**Shashank Vatedka** *                                                      *shashankvatedka@ee.iith.ac.in*
*Department of Electrical Engineering*
*Indian Institute of Technology Hyderabad, India*

**Reviewed on OpenReview:** *https://openreview.net/forum?id=g95C4zIEPg*

## Abstract

In this work, we study the problem of distributed mean estimation with 1-bit communication constraints when the variance is unknown. We focus on the setting where each user has access to one i.i.d. sample drawn from a distribution belonging to a *location–scale family*, and is limited to sending just a single bit of information to a central server whose goal is to estimate the mean. We propose simple non-adaptive and adaptive protocols and show that both achieve asymptotic normality. We derive bounds on the asymptotic (in the number of users) Mean Squared Error (MSE) achieved by these protocols. For a class of symmetric log-concave distributions, we derive matching lower bounds for the MSE of adaptive protocols, establishing the optimality of our scheme. Furthermore, we develop a lower bound on the MSE for non-adaptive protocols that applies to any symmetric strictly log-concave distribution, using a refined squared Hellinger distance analysis. Through this, we show that for many common distributions, including a subclass of the generalized Gaussian family, the asymptotic minimax MSE achieved by the best non-adaptive protocol is strictly larger than that achieved by our simple adaptive protocol. We also demonstrate that increasing the number of bits per user can only marginally reduce the asymptotic MSE of adaptive protocols. Our simulation results confirm a positive gap between the adaptive and non-adaptive settings, aligning with the theoretical bounds.

## 1 Introduction

The rise of big data has made distributed data analysis essential, as large datasets are often spread across multiple locations. Traditional data processing and learning techniques are often insufficient in such scenarios, motivating advances in distributed learning and estimation (McMahan et al., 2017; Chen et al., 2021; Kairouz et al., 2021). In many problems such as distributed optimization and federated learning, communication often emerges as the main bottleneck, creating the need to develop communication-efficient algorithms for distributed learning, estimation, and optimization (Alistarh et al., 2017; Chen et al., 2021; Li et al., 2020; Bernstein et al., 2018; Kairouz et al., 2021; Chen et al., 2018; Wang et al., 2018).

Extreme communication efficiency is also crucial in many signal processing applications, including IoT and sensor networks, where we typically have tens to hundreds of low-power devices that make noisy measurements which must be communicated to a fusion center for subsequent decisions or estimation. In such scenarios, severe power and bandwidth constraints have motivated substantial recent work on understand-

---

*The work of Shashank Vatedka was supported by a Core Research Grant (CRG/2022/004464) from the Anusandhan National Research Foundation (ANRF).

ing the trade-off between communication cost and estimation accuracy (Heinzelman et al., 2000; Ribeiro & Giannakis, 2006; Luo, 2005; Ben-Basat et al., 2024).

In particular, there has been growing interest in distributed algorithms for machine learning, optimization, and statistical inference when each of a large number of users is allowed to communicate only one bit per sample (Seide et al., 2014; Zhu et al., 2021; Fan et al., 2022; Tang et al., 2021). While such requirements may seem extreme, they offer significant benefits in terms of system design, particularly when using low-cost, resource-constrained devices.

There is a large body of recent work on distributed mean estimation (DME) under such communication constraints. DME in the high-dimensional setting forms a basic building block for several distributed learning and optimization algorithms (Suresh et al., 2017; Amiraz et al., 2022; Liu et al., 2023).

On a similar vein, there has been significant progress in understanding the fundamental limits of parametric estimation in location families—particularly Gaussian (Cai & Wei, 2022b; 2024) and symmetric log-concave (Kipnis & Duchi, 2022; Kumar & Vatedka, 2023) location families—as well as non-parametric density (Acharya et al., 2023) and mean (Lau & Scarlett, 2025) estimation under communication constraints. Estimation with more general information constraints has been studied in (Acharya et al., 2020a;b; 2021; Koltchinskii et al., 2023), while lower bounds on minimax risk for distributed statistical estimation (Zhang et al., 2013) and communication-efficient algorithms for distributed optimization on large-scale data (Zhang et al., 2012) have been established. Practical algorithms have been proposed in (Battey et al., 2018; Deisenroth & Ng, 2015). The distributed setting with one-bit information sharing is particularly challenging and has been extensively studied (Venkitasubramaniam et al., 2007; Chen & Varshney, 2009). More recently, (Han et al., 2021; Barnes et al., 2020; Rahmani et al., 2024; Shah et al., 2025) derived lower bounds for distribution learning in parametric and nonparametric settings. The framework applies to both fixed and sequential message-passing protocols for various problems including communication-constrained mean and covariance estimation for Gaussian distributions. Understanding the parametric setting can yield new insights and aid the design of algorithms for problems such as distributed optimization and federated learning with communication constraints.

Location-scale models have been widely studied in statistics, economics, and sociology. In economics, such models form the basis of price, demand, and welfare estimation, from classical discrete-choice models Deaton & Muellbauer (1980); McFadden (1974) to more recent work addressing arbitrage-free forecasting and heavy-tailed market behavior Carvalho & Hahn (2018); Guggenberger & Jansson (2022). In quantitative finance, location–scale assumptions underpin models of returns and volatility, including ARCH/GARCH and their modern Bayesian and data-driven extensions Bollerslev (1987); Campbell et al. (1997); Yu & Meyer (2019); Sirignano & Cont (2019). Similarly, in sociology and psychometrics, latent-trait models such as the Rasch and logistic item-response formulations depends on scale–location structures Rasch (1960); Birnbaum (1968); Jeon & Rabe-Hesketh (2019).

In many of these problems, data is often generated across heterogeneous, geographically dispersed, or privacy-sensitive sources-retail outlets, financial exchanges, or survey respondents—where transmitting full-resolution observations is impractical due to bandwidth, latency, or confidentiality constraints Suresh et al. (2017); Konečný et al. (2016); Duchi et al. (2018). In such settings, one-bit and low-bit communication primitives could provide an attractive alternative by reducing communication, provide privacy (with additional differentially private mechanisms), and enable lightweight protocols that aggregate information without requiring centralized raw data access Zhang et al. (2013); Braverman et al. (2016); Dana & Duchi (2022). Thus, the one-bit estimators developed in this work could potentially have applications in many of these areas as well.

## 1.1 Contributions

In this paper, we consider distributed mean estimation in a setting where each user has a single i.i.d. sample and is allowed to share only one bit of information with a central server. Many of the existing works on DME, including (Kipnis & Duchi, 2022; Cai & Wei, 2024; Kumar & Vatedka, 2023), assume that the mean is the only unknown parameter, which may not hold in practice. We consider scale-location families, where both the mean and variance are unknown, and the variance is treated as a nuisance parameter. We design

simple non-adaptive and adaptive protocols for this problem, and derive bounds on the asymptotic mean squared error (MSE) achieved by our protocols.

Despite the recent surge of interest in such problems, much less is known about the performance gap between adaptive and non-adaptive protocols when both the mean and variance are unknown. For many communication-constrained estimation/testing problems, it is an interesting open question as to whether non-adaptive protocols are strictly suboptimal compared to adaptive protocols.

The main contribution of our work is to show that for a broad class of distributions, including the hyperbolic secant and generalized Gaussian with shape parameter $\beta < 1.85$, the asymptotic minimax MSE achieved by the best non-adaptive estimator is provably higher than that achieved by our simple 2-round adaptive protocol. We do so by deriving a general lower bound on the minimax MSE of non-adaptive estimators for arbitrary symmetric strictly log-concave distributions. We then compare this against the MSE achieved by our adaptive protocol. We also derive a lower bound on the MSE achieved by adaptive estimators and prove that for a broad class of distributions, our simple 2-round protocol is in fact optimal. We also also compare our results with a lower bound on asymptotic MSE for the case where no communication constraints are imposed on the users, and observe that for many distributions our adaptive protocol achieves an MSE that is close to this lower bound.

Although we mainly study this problem through the lens of parametric estimation, the algorithms and insights derived here could aid the design of improved algorithms for distributed optimization and federated learning when there are severe communication constraints.

A short recorded video is provided as supplementary material here Kumar & Vatedka (2026a).

## 2 Problem Setup and Summary of Results

### 2.1 Problem Setup

**Definition 2.1.** Let $f_X$ be a probability density function (pdf) on $\mathbb{R}$ with zero mean and unit variance, and let $F_X$ be the corresponding cumulative distribution function (cdf).

The *location family* associated with $f_X$ is defined as:

$$\mathcal{L}(f_X) \triangleq \{f_{X,\mu}(x) = f_X(x - \mu) : \ \mu \in \mathbb{R}\}. \tag{1}$$

The *scale-location family* associated with $f_X$ is defined as:

$$\mathcal{L}_{\mathrm{SL}}(f_X) \triangleq \left\{ f_{X,\mu,\sigma}(x) = \frac{1}{\sigma} f_X\left(\frac{x - \mu}{\sigma}\right) : \ \mu \in \mathbb{R}, \ \sigma > 0 \right\}. \tag{2}$$

Throughout the paper, we assume that $f_X$ is non-zero, $F_X$ is invertible, and $F_X^{-1}$ is differentiable at all points in $\mathbb{R}$.

The main goal of this paper is to establish fundamental bounds on the achievable mean squared error (MSE) for estimating $\mu \in \mathbb{R}$ in a distributed setting when $\sigma > 0$ is unknown and treated as a nuisance parameter.

Let us formally describe the problem. Suppose $X_1, X_2, \ldots, X_n$ are i.i.d. samples drawn from a distribution with density function $f_{X,\mu,\sigma}$ belonging to $\mathcal{L}_{\mathrm{SL}}(f_X)$. We consider a central server and $n$ users, where user $i$ has access to only one sample $X_i$ (also called the $i$'th input/source) and is allowed to send a single bit of information $Y_i \in \{0, 1\}$ to the server.

We assume that $f_X$ is exactly known to the server, while $(\mu, \sigma) \in \mathbb{R} \times \mathbb{R}_{>0}$ are unknown. The goal is to design a communication protocol $\Pi$ that enables the server to reliably estimate $\mu$ from $Y_1, \ldots, Y_n$.

Designing $\Pi$ amounts to specifying $n$ encoders (one for each user) that map $X_i$ to $Y_i$, and a decoder (at the server) that maps $(Y_1, \ldots, Y_n)$ to an estimate $\hat{\mu}$ of $\mu$. Importantly, we do not restrict the encoders to be identical, i.e., we can design a different map for each user if required.

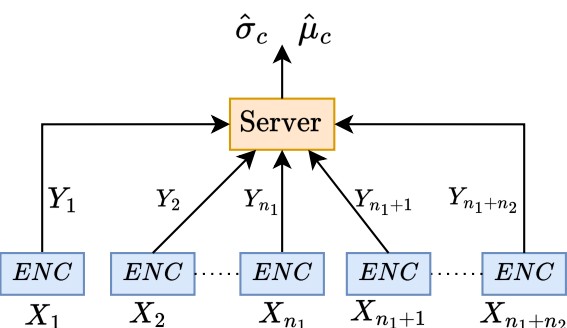

Figure 1: Illustration of our non-adaptive protocol. We partition the $n$ users into subsets of $n_1$ and $n_2$ users. Each user independently encodes their sample using a 1-bit quantizer, with threshold $\theta_1$ for the first $n_1$ users and $\theta_2$ for the remaining $n_2$ users.

We call a protocol *non-adaptive* if each $Y_i$ depends only on $X_i$; it is *adaptive* if $Y_i$ can depend on $X_i$ as well as $Y_1, \ldots, Y_{i-1}$.

The performance of a protocol is quantified via the asymptotic mean squared error:

$$\lim_{n \to \infty} n \cdot \mathrm{MSE}(\hat{\mu}) \triangleq \lim_{n \to \infty} n \cdot \mathbb{E}\left[|\mu - \hat{\mu}|^2\right], \tag{3}$$

which may, in general, depend on $\mu$ and $\sigma$.

In this work, we consider both adaptive and non-adaptive protocols for scale-location families with symmetric log-concave base density $f_X$. We derive bounds on the optimal asymptotic MSE in these settings and show that for many distributions of interest, there is a positive and quantifiable gap between the optimal adaptive and non-adaptive protocols.

### 2.1.1 Notation

For a sequence of random variables $X_1, X_2, \ldots$ and a random variable $X$ all defined over the same probability space, we denote $X_n \xrightarrow{\text{a.s.}} X$ if the sequence of random variables converges almost surely to $X$. Likewise, $X_n \xrightarrow{d} X$ if the sequence converges in distribution to $X$.

We say that a protocol is strongly consistent if the final estimate $\hat{\mu} \xrightarrow{\text{a.s.}} \mu$ as $n \to \infty$ for every $\mu, \sigma$. Similarly, we say that $\hat{\mu}$ is asymptotically normal if $\sqrt{n}(\hat{\mu} - \mu)$ converges in distribution to a Gaussian as $n \to \infty$ for every $\mu, \sigma$.

## 2.2 Summary of Results

We first study a simple threshold-based 1-bit non-adaptive protocol that yields strongly consistent and asymptotically normal estimates for jointly estimating both $\mu$ and $\sigma$. In Theorem 3.1, we derive the achievable MSE for estimating $\mu$ and $\sigma$ using this scheme.

We then derive a lower bound on the asymptotic MSE for non-adaptive protocols, and this is one of the main contributions of this work. Specifically, we show that if $\log \frac{1}{f_X}$ is strictly convex, then for any non-adaptive protocol,

$$\lim_{n \to \infty} \sup_{\mu} \left(n \cdot \mathrm{MSE}(\hat{\mu})\right) \geqslant \frac{0.1034\,\sigma^2}{T(f_X)}, \tag{4}$$

where $T(f_X)$ is a parameter that depends on $f_X$. See Theorem 3.2 for more details.

We also study an intuitive multi-threshold non-adaptive protocol, where each user is assigned a different threshold. In Theorem 3.3, we establish bounds on the bias and variance of the resulting estimator.

We then design a simple adaptive protocol for estimating $\mu$. This requires minimal adaptation and can be summarized at a high level as follows: We first run our non-adaptive protocol using only the first $n' = o(n)$

users, and the server produces coarse estimates $(\hat{\mu}_c, \hat{\sigma}_c)$ of the mean and variance respectively. The estimate $\hat{\mu}_c$ is broadcast to the remaining $n - n'$ users. These users then independently encode their samples to produce $Y_{n'+1}, \ldots, Y_n$, which are used by the server to produce the final estimate $\hat{\mu}$. In Theorem 4.1, we show that this simple adaptive scheme is strongly consistent, asymptotically normal, and achieves

$$\lim_{n \to \infty} n \cdot \text{MSE}(\hat{\mu}) = \frac{\sigma^2}{4 f_X^2(0)}. \tag{5}$$

We then show that this is optimal for a large subclass of symmetric, strictly log-concave distributions, by proving a matching lower bound on the MSE of any adaptive protocol (Theorem 4.3). As a consequence of our results, we are able to show the strict suboptimality of non-adaptive protocols compared to adaptive protocols. For several important examples—including generalized Gaussian distributions (GGD):

$$f_X(x) = \frac{\beta}{2\alpha \Gamma(1/\beta)} \exp\left(-(|x|/\alpha)^\beta\right), \qquad \alpha \triangleq \sqrt{\frac{\Gamma(1/\beta)}{\Gamma(3/\beta)}}, \tag{6}$$

with $1 < \beta < 1.85$, and the hyperbolic secant—we show that the lower bound in equation 4 is strictly larger than the $\frac{\sigma^2}{4 f_X^2(0)}$ achieved by our simple adaptive protocol, thereby establishing a positive and quantifiable gap between the optimal adaptive and non-adaptive performance.

We compare these bounds with one obtained using the Fisher information, corresponding to the centralized setting (equivalently, where there are no communication constraints). For many distributions, our simple adaptive protocol achieves an asymptotic MSE that is close to this lower bound, which shows that increasing the number of bits communicated per user can only marginally improve the performance.

We finally conclude with some comments and open questions.

### 2.3  Related Work

Perhaps the closest work to this paper is (Kipnis & Duchi, 2022), which studied 1-bit DME for location families, where the mean is the only unknown parameter. Our protocols are inspired by their schemes for location families, and we also borrow some high-level ideas in calculating the asymptotic MSE. However, their work assumes that the mean is the only unknown parameter, and their results and techniques do not directly extend to the case where the variance is unknown. The analysis of our adaptive protocol is also significantly more challenging than theirs, as we must also obtain a coarse estimate of the variance, which introduces additional dependencies in the MSE derivation. Moreover, we take a totally different approach towards lower bounding the minimax MSE for non-adaptive protocols.

The works (Cai & Wei, 2022a; 2024) studied DME of Gaussian distributions under more general multi-bit communication constraints, and derived bounds on the minimax MSE for finite (but large) $n$. However, both works assume that the range of $\mu$ is bounded (e.g., $\mu \in [0, 1]$), and the results do not extend if this assumption is relaxed. While (Cai & Wei, 2024) assumes that the variance is known exactly, (Cai & Wei, 2022a) assumes that the variance is unknown but requires the protocol to know a lower bound on $\sigma$ in advance (with the achievable MSE depending on this bound). In fact, (Cai & Wei, 2022a) prove a gap between non-adaptive and adaptive protocols for Gaussian scale-location families. However, their approaches are tailored specifically to the Gaussian setting with restrictions on the range of $\mu, \sigma$, their protocols require more than one bit per user, and their methods do not extend easily to more general distributions. In contrast, we make no boundedness assumptions on $\mu$ or $\sigma$, and our results hold for a broad class of symmetric, strictly log-concave distributions.

It is worth noting that (Kipnis & Duchi, 2022) claimed strict suboptimality of non-adaptive protocols when the mean is the only unknown parameter. However, their lower bound for non-adaptive protocols relies on restrictive assumptions (see (Kipnis & Duchi, 2022, Assumption A2)), and it is possible to design schemes that violate these. Indeed, (Cai & Wei, 2024) provide such a non-adaptive protocol for the Gaussian location family that achieves order-optimal MSE even under communication constraints. In our work, we prove strict suboptimality of non-adaptive protocols in the $\sigma$-unknown setting for a broad class of symmetric, strictly log-concave distributions, thereby generalizing the Gaussian-specific results of (Cai & Wei, 2022a).

Very recently, (Shah et al., 2025) analyzed properties of the maximum likelihood (ML) estimator using threshold-based non-adaptive protocols for parametric estimation of a class of exponential families. Specifically, they study protocols where $Y_i = \mathbf{1}_{\{X_i \geqslant \tau_i\}}$, where $\mathbf{1}_{\{\cdot\}}$ denotes the indicator function and for $i = 1, 2, \ldots, n$, $\tau_i \in \mathbb{R}$ is a predefined threshold. The authors show that the ML estimator is consistent and asymptotically normal. Using this, they are able to compute the asymptotic mean squared error for Gaussian scale-location families as a function of $\tau_1, \tau_2, \ldots$, but further optimization of the thresholds and deriving simple closed-form upper/lower bounds was left for future work. In comparison, we design non-adaptive and adaptive protocols for mean estimation over general scale-location families, derive closed-form achievable and converse bounds, and prove matching lower bounds for adaptive protocols over symmetric, strictly log-concave distributions. Our lower bound for non-adaptive protocols holds in the broadest sense (subsuming threshold-based protocols).

More recently, (Lau & Scarlett, 2025) studied nonparametric estimation of the mean with 1-bit communication constraints, but under a probably approximately correct (PAC) framework. They derived upper and lower bounds on the sample complexity, and notably showed that a two-stage adaptive protocol is order optimal. In contrast, we derive bounds on the asymptotic minimax MSE, with an aim to characterize the exact constants.

A number of works have derived lower bounds for various estimation problems with communication constraints (Han et al., 2021; Zhang et al., 2012), (Zhang et al., 2013), (Barnes et al., 2019; 2020). However, these results do not directly yield lower bounds for our problem. Many of these works employ either the Van Trees inequality or Le Cam's method (Van Trees, 2004; Gill & Levit, 1995; Le Cam, 2012; Polyanskiy & Wu, 2024), and we use both approaches in deriving our bounds. Our adaptive lower bound uses the Van Trees inequality, similar to (Kipnis & Duchi, 2022), while our non-adaptive lower bound is obtained by upper-bounding the squared Hellinger distance between transcripts under two different mean values, followed by an application of Le Cam's method (Le Cam, 1973; Polyanskiy & Wu, 2024).

A parallel line of work (e.g., (Konečný & Richtárik, 2018; Mayekar et al., 2021; Ben-Basat et al., 2024; Babu et al., 2025)) studied distributed *empirical mean estimation* for high-dimensional vectors, with applications to distributed optimization and federated learning. Unlike our work where the samples are assumed to be i.i.d., here the samples are fixed but arbitrarily chosen (typically to be of unit norm). The goal is to design communication-efficient compressors and sketches and study accuracy–communication tradeoffs. While the setup is slightly different, many high-level ideas between these works and ours (e.g., randomized quantizers and sketching ideas) are conceptually related.

Along similar lines, Vardhan & Mazumdar (2025) have recently investigated communication-efficient distributed mean estimation with collaborative one-bit compressors, in which clients use private and shared randomness to compress their source. Their basic scheme operates by deliberately perturbing the source with independent gaussian noise, and the server uses a mean estimator that looks similar to the one described in Section 3.1. The authors also extend this to high-dimensional sources and derive bounds on the worst-case MSE assuming that the input samples lie within a (known) bounded set.

## 3 Non-Adaptive Protocols

### 3.1 A Consistent Protocol for Location Families

To provide some intuition for the design of our protocols, let us begin with the simple non-adaptive protocol considered in (Kipnis & Duchi, 2022) for estimating location families. For now, assume that each $X_i$ is an i.i.d. sample drawn according to $f_{X,\mu}(x) = f_X(x - \mu)$, where $\mu \in \mathbb{R}$ is the only unknown parameter.

User $i$ transmits

$$Y_i = \mathbb{1}_{\{X_i < \theta\}},$$

where $\theta \in \mathbb{R}$ is a pre-set threshold and $\mathbb{1}_E$ is the indicator of event $E$. At the server, we compute $\bar{Y}_n$ as follows:

$$\bar{Y}_n = \frac{1}{n} \sum_{i=1}^{n} Y_i \xrightarrow{\text{a.s.}} F_X(\theta - \mu), \tag{7}$$

where convergence follows from the strong law of large numbers. Using this, the server can estimate $\mu$ by inverting the quantile (Kipnis & Duchi, 2022; Lehmann & Casella, 2006) as follows

$$\hat{\mu} = \theta - F_X^{-1}(\bar{Y}_n).$$

From the continuous mapping theorem (Casella & Berger, 2002), $\hat{\mu} \xrightarrow{\text{a.s.}} \mu$ as $n \to \infty$, and hence $\hat{\mu}$ is a strongly consistent protocol of $\mu$.

As discussed in (Kipnis & Duchi, 2022), $\sqrt{n}(\bar{Y}_n - F_X(\theta - \mu))$ converges in distribution to a Gaussian with zero mean and variance $F_X(\theta - \mu)(1 - F_X(\theta - \mu))$ as $n \to \infty$. Using the delta method, they show that $\sqrt{n}(\hat{\mu} - \mu)$ is asymptotically normal with

$$\lim_{n \to \infty} n \cdot \text{MSE}(\hat{\mu}) = \frac{F_X(\theta - \mu)(1 - F_X(\theta - \mu))}{f_X^2(\theta - \mu)}.$$

### 3.2 A Consistent Non-Adaptive Protocol for Scale-Location Families

As a warmup, let us extend the ideas above to design a 1-bit non-adaptive protocol for estimating $\mu, \sigma$.

We partition the users into two subsets of size $n_1$ and $n_2$ respectively. Here[1] $n_1 = K_1 n$ and $n_2 = K_2 n$, where $K_1$ and $K_2$ are constants independent of $n$, and $K_1 + K_2 = 1$. We select two different thresholds $\theta_1$ and $\theta_2$, and the encoders operate as follows:
For $1 \leqslant i \leqslant n_1$,

$$Y_i = \begin{cases} 1 & \text{if } X_i < \theta_1, \\ 0 & \text{otherwise.} \end{cases}$$

For $n_1 + 1 \leqslant i \leqslant n$,

$$Y_i = \begin{cases} 1 & \text{if } X_i < \theta_2, \\ 0 & \text{otherwise.} \end{cases}$$

The users share the encoded samples with server. The server first computes

$$F_1 \triangleq \frac{1}{n_1} \sum_{i=1}^{n_1} Y_i, \qquad F_2 \triangleq \frac{1}{n_2} \sum_{i=n_1+1}^{n_2} Y_i,$$

and then

$$\alpha_1 \triangleq F_X^{-1}(F_1) \quad \text{and} \quad \alpha_2 \triangleq F_X^{-1}(F_2). \tag{8}$$

By the strong law of large numbers, $F_1$ and $F_2$ almost surely converge to $F_X\left(\frac{\theta_1 - \mu}{\sigma}\right)$ and $F_X\left(\frac{\theta_2 - \mu}{\sigma}\right)$ respectively. Now, we define the estimates of $\sigma$ and $\mu$ as follows:

$$\hat{\sigma}_c \triangleq \frac{\theta_1 - \theta_2}{\alpha_1 - \alpha_2} \xrightarrow{\text{a.s.}} \sigma \text{ and } \hat{\mu}_c \triangleq \frac{\alpha_1 \theta_2 - \alpha_2 \theta_1}{\alpha_1 - \alpha_2} \xrightarrow{\text{a.s.}} \mu, \tag{9}$$

where convergence follows from the continuous mapping theorem (Casella & Berger, 2002, Sec. 5.5). With this setup, the MSE achieved by this protocol is presented below.

---

[1]To keep the calculations simple, we assume that $K_1 n, K_2 n$ are integers. However, the results do not change if we choose $n_1, n_2$ to be the closest integers to $K_1 n, K_2 n$ respectively.

**Theorem 3.1.** *For every $\mu \in \mathbb{R}$ and $\sigma > 0$, the protocol in Sec. 3 is strongly consistent, asymptotically normal, and satisfies*

$$\lim_{n\to\infty} n \cdot \mathrm{MSE}(\hat{\mu}_c) = \frac{\sigma^2}{(\theta_1 - \theta_2)^2} \left[ (\theta_2 - \mu)^2 \frac{K_1 \sigma_1^2}{f_X^2\left(\frac{\theta_1 - \mu}{\sigma}\right)} + (\theta_1 - \mu)^2 \frac{K_2 \sigma_2^2}{f_X^2\left(\frac{\theta_2 - \mu}{\sigma}\right)} \right],$$

$$\lim_{n\to\infty} n \cdot \mathrm{MSE}(\hat{\sigma}_c) = \frac{\sigma^4}{(\theta_1 - \theta_2)^2} \left[ \frac{K_1 \sigma_1^2}{f_X^2\left(\frac{\theta_1 - \mu}{\sigma}\right)} + \frac{K_2 \sigma_2^2}{f_X^2\left(\frac{\theta_2 - \mu}{\sigma}\right)} \right],$$

*where for $i = 1, 2$,*

$$\sigma_i^2 \triangleq F_X\left(\frac{\theta_i - \mu}{\sigma}\right)\left(1 - F_X\left(\frac{\theta_i - \mu}{\sigma}\right)\right).$$

See Appendix A for the proof.

We do not claim that the above protocol is optimal. A careful observation reveals that the asymptotic value of $n \cdot \mathrm{MSE}(\hat{\mu}_c)$ and $n \cdot \mathrm{MSE}(\hat{\sigma}_c)$ depend on $\mu$, and that for fixed $\theta_1, \theta_2$, this grows unbounded as $\mu \to \infty$ or $\mu \to -\infty$. However, we will use this as a basic building block for our adaptive scheme. While it yields strongly consistent estimates for both $\mu$ and $\sigma$, its performance can be significantly improved when a small amount of feedback can be provided to a subset of the users.

### 3.3 Lower Bound for Non-Adaptive Protocols

We now present a general lower bound on the performance of non-adaptive one-bit protocols.

**Theorem 3.2.** *Assume that $f_X(x) = e^{-\phi(x)}$ where $\phi(x)$ is symmetric, differentiable, strictly convex, and is upper bounded by a polynomial[2]. Then, for every 1-bit non-adaptive protocol,*

$$\sup_{\mu} \lim_{n\to\infty} n \cdot \mathbb{E}\left[(\hat{\mu} - \mu)^2\right] \geq \frac{\alpha^\star \sigma^2}{T(f_X)} \tag{10}$$

*where*

$$T(f_X) \triangleq \int_0^{h^*} \phi'\left(h^{-1}(t)\right) h^{-1}(t)\, \mathrm{d}t,$$

$$h(x) \triangleq 2\phi'(x) f_X(x), \qquad h^* \triangleq \max_{x \geq 0} h(x),$$

*are quantities that only depend on $f_X$, and $h^{-1} : [0, h^*] \to \mathbb{R}$. Here*

$$\alpha^\star \triangleq \max_{t \geq 0}\ t\left(1 - \sqrt{1 - e^{-2t}}\right) \approx 0.1034.$$

*Proof.* We use Le Cam's two-point method by identifying a suitable hypothesis testing problem and lower bounding the MSE in terms of the total variation distance between distributions of the transcript corresponding to the two hypotheses for any non-adaptive protocol.

Specifically, fix $\mu \in \mathbb{R}$ and $\sigma > 0$, choose $\epsilon > 0$ (possibly depending on $n$), and define the symmetric alternatives

$$\mu_- := \mu - \epsilon\sigma, \qquad \mu_+ := \mu + \epsilon\sigma. \tag{11}$$

Each user obtains an i.i.d. sample $X_i$ drawn according to one of two distributions based on the true hypothesis:

$$\mathcal{H}_- :\ X_i \sim f_{X,\mu_-,\sigma}(\cdot), \qquad \mathcal{H}_+ :\ X_i \sim f_{X,\mu_+,\sigma}(\cdot).$$

Let $P_{Y^n}^-$ and $P_{Y^n}^+$ be the distributions of the encoded sequence $Y^n = [Y_1, \ldots, Y_n]$ for any given non-adaptive protocol corresponding to the two hypotheses.

---

[2]We only require that the degree of the polynomial be a constant, independent of $n$.

Using Le Cam's method (Wu, 2020, Theorem 10.2),

$$\sup_{\mu \in \{\mu_-, \mu_+\}} \mathbb{E}\left[(\hat{\mu} - \mu)^2\right] \geqslant \frac{(\mu_+ - \mu_-)^2}{4} \left(1 - \mathrm{TV}(P_{Y^n}^-, P_{Y^n}^+)\right) \tag{12}$$

$$= \epsilon^2 \sigma^2 \left(1 - \mathrm{TV}(P_{Y^n}^-, P_{Y^n}^+)\right), \tag{13}$$

where TV denotes the total variation distance between the two joint distributions.

Let[3]

$$H^2(P_{Y^n}^+, P_{Y^n}^-) \triangleq \frac{1}{2} \sum_{y^n \in \{0,1\}^n} \left(\sqrt{P_{Y^n}^+(y^n)} - \sqrt{P_{Y^n}^-(y^n)}\right)^2$$

denote the squared Hellinger distance between the two distributions.

For a non-adaptive protocol, $Y_1, \ldots, Y_n$ are independent (but need not be identically distributed). Hence the Bhattacharyya coefficients factorize (Wright & Tang, 2024, Proposition 1.4):

$$\rho(P_{Y^n}^+, P_{Y^n}^-) = \prod_{i=1}^n \rho(P_{Y_i}^+, P_{Y_i}^-).$$

From Theorem B.1, for each $i$ we have

$$H^2(P_{Y_i}^+, P_{Y_i}^-) \leqslant \epsilon^2 \big(T(f_X) + o(1)\big),$$

so

$$\rho(P_{Y_i}^+, P_{Y_i}^-) \geqslant 1 - \epsilon^2 \big(T(f_X) + o(1)\big).$$

Therefore,

$$\rho(P_{Y^n}^+, P_{Y^n}^-) \geqslant (1 - \epsilon^2 T(f_X) + o(\epsilon^2))^n = e^{-n\epsilon^2 T(f_X) + o(1)}.$$

By the standard inequality relating total variation and the Bhattacharyya coefficient (see, e.g., (Tsybakov, 2009, cf. Lemma 2.6); also the derivation of the Bretagnolle–Huber bound (Bretagnolle & Huber, 1979)), we have

$$\mathrm{TV}(P_{Y^n}^-, P_{Y^n}^+) \leqslant \sqrt{1 - \rho(P_{Y^n}^+, P_{Y^n}^-)^2} \leqslant \sqrt{1 - e^{-2t + o(1)}},$$

where $t = n\epsilon^2 T(f_X)$. Therefore,

$$n \cdot \sup_{\mu \in \{\mu_-, \mu_+\}} \mathbb{E}\big[(\hat{\mu} - \mu)^2\big] \geqslant \frac{\sigma^2}{T(f_X)} \left(t\left(1 - \sqrt{1 - e^{-2t}}\right) + o(1)\right).$$

Maximizing the right-hand side over $t \geqslant 0$ yields the constant $\alpha^\star$, which is approximately 0.10340 and $o(1) \to 0$ as $n \to \infty$. This completes the proof. $\qquad\square$

### 3.4 Multi-Threshold Non-Adaptive Protocol for Non-Parametric Mean Estimation

We now provide a natural multi-threshold protocol that works for general distributions (not just scale-location families), yielding a non-parametric estimator for the mean.

Assuming $\mathbb{E}[|X|] < \infty$, the expectation of $X$ can be written as

$$\mathbb{E}[X] = \int_0^\infty \mathbb{P}(X > z)\, \mathrm{d}z - \int_{-\infty}^0 \mathbb{P}(X < z)\, \mathrm{d}z \tag{14}$$

Define a partition of $\mathbb{R}$ as

$$\theta_{-m} < \theta_{-m+1} < \cdots < \theta_{-1} < \theta_0 < \theta_1 < \cdots < \theta_m,$$

---

[3]Note that we have a factor of 1/2 in the definition of the squared Hellinger distance, which is absent in many standard texts.

with $\theta_0 = 0$, and

$$\theta_j - \theta_{j-1} = \Delta \text{ for all } j \in \{-m+1, \ldots, m\}$$

for some parameter $\Delta > 0$. In particular, we choose the parameters so that $\theta_m = -\theta_{-m} \to \infty$ and $\Delta \to 0$ as $n \to \infty$.

Define the midpoints

$$\bar{\theta}_j = \frac{\theta_j + \theta_{j+1}}{2}, \quad j \in \{-m, \ldots, -1\},$$

and

$$\bar{\theta}_j = \frac{\theta_j + \theta_{j-1}}{2}, \quad j \in \{1 \ldots, m\},$$

A natural idea is to approximate equation 14 by a Riemann sum:

$$\mathbb{E}[X] \approx \hat{\mu}_\Theta \triangleq \sum_{j=1}^{m} \mathbb{P}(X > \bar{\theta}_j)(\theta_j - \theta_{j-1}) - \sum_{j=-m}^{-1} \mathbb{P}(X < \bar{\theta}_j)(\theta_{j+1} - \theta_j). \tag{15}$$

### 3.4.1 The Multi-Threshold Protocol

Our protocol can be described as follows: We partition the $n$ users into groups of $K = n/(2m)$ users each. For convenience, let us index the users by $(i, j)$, where $i \in \{1, 2, \ldots, K\}$ labels the group and $j \in \{-m, \ldots, -1, 1, \ldots, m\}$ indexes a user within the group.

The $(i, j)$'th user sends

$$Y_{i,j} = \begin{cases} 1_{\{X_{i,j} < \bar{\theta}_j\}}, & \text{if } j < 0 \\ 1_{\{X_{i,j} > \bar{\theta}_j\}}, & \text{if } j > 0 \end{cases}$$

and the server forms empirical averages

$$\bar{I}_j = \frac{1}{K} \sum_{i=1}^{K} Y_{i,j}, \qquad \text{for } j \in \{-m, \ldots, -1, 1, \ldots, m\}$$

The server outputs

$$\hat{\mu}_{MT} = \sum_{j=1}^{m} \bar{I}_j \Delta - \sum_{j=-m}^{-1} \bar{I}_j \Delta \tag{3}$$

**Theorem 3.3.** *Let $X$ be any random variable with $\mathbb{E}[|X|] < \infty$, distribution function $F_X$, and bounded density $f_X \leq L$. Consider the multi-threshold protocol defined above. For every $\mu \in \mathbb{R}$ and $\sigma > 0$,*

$$|\mathbb{E}[\hat{\mu}_{MT}] - \mathbb{E}[X]| \leq \frac{Lm\Delta^2}{\sigma}.$$

*If we choose the parameters such that $\Delta \to 0$ and $\theta_m = -\theta_{-m} \to \infty$, then*

$$\text{Var}(\hat{\mu}_{MT}) = \frac{\Delta}{K} \left( \int_{-\infty}^{\infty} F_X(x)(1 - F_X(x)) \mathrm{d}x + o(1) \right)$$

*provided that the above integral exists.*

The proof is provided in Appendix E.

Observe that

$$\frac{\Delta}{K} = \frac{(\theta_m + \theta_{-m})}{2mK} = \frac{(\theta_m - \theta_{-m})}{2n}.$$

Therefore, if we require $\theta_m = -\theta_{-m} \to \infty$, then we cannot have the variance of $\hat{\mu}_{MT} = O(1/n)$. However, the careful reader may note that if $X$ is bounded, then this requirement is waived, and we can indeed obtain MSE that decays as $O(1/n)$.

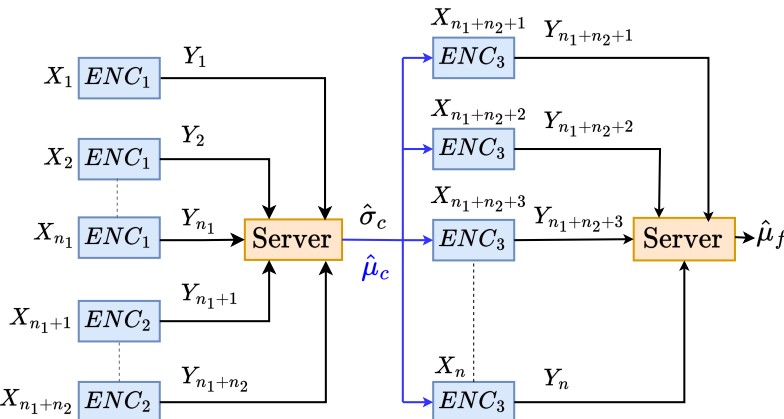

Figure 2: Our two-round adaptive protocol: The first $n_1 + n_2$ users use the non-adaptive scheme to produce coarse estimates $\hat{\mu}_c, \hat{\sigma}_c$. The value of $\hat{\mu}_c$ is broadcast to the remaining $n_3$ users, who then send 1-bit messages using $\hat{\mu}_c$ as their quantization threshold.

This illustrates that nonparametric estimation of the mean can be more challenging. From Theorem 3.1, we see that for a scale-location family, we can do significantly better than the above even with just two thresholds. However, the above theorem holds even when the mean and variance are not the only unknown parameters.

Please note that Section 3.4 develops a multi-threshold non-adaptive estimator as a general-purpose, non-parametric construction. As discussed in that , such estimators may not achieve an $O(1/n)$ MSE unless additional conditions (such as boundedness of $X$) are imposed, because the variance term scales with the range of the threshold grid. In contrast, for location–scale families, Theorem 3.1 shows that the two-threshold estimator of Section 3.2 already achieves significantly better performance using only a pair of carefully chosen thresholds. For this reason, the adaptive scheme in Section 4 builds exclusively on the two-threshold protocol of Section 3.2, while the multi-threshold estimator of Section 3.4 is included only as a general extension and is not used in the adaptive procedure. All references to *the non-adaptive protocol* in what follows should therefore be understood as referring to the method of Section 3.2.

## 4 Adaptive Protocol

### 4.1 Simple Two-Round Adaptive Protocol

We now describe a simple adaptive scheme that can be implemented with two rounds of communication[4]. We will subsequently show that for many common distributions, this outperforms the best non-adaptive estimator.

We partition the $n$ users into three disjoint subsets of sizes $n_1$, $n_2$, and $n_3$, where $n_1 = n_2 = n_3/(\log n_3)$ and $n_1 + n_2 + n_3 = n$. The first $n_1 + n_2$ users participate in the *first round*, while the remaining $n_3$ users participate in the *second round*. The protocol is illustrated in Fig. 2.

**First round:** We run the non-adaptive protocol of Sec. 3.2 using the first $n_1 + n_2$ users to obtain coarse estimates $\hat{\mu}_c$ and $\hat{\sigma}_c$. The estimate $\hat{\mu}_c$ is then broadcast to the remaining $n_3$ users.[5]

**Second round:** Each user $i$ with $n_1 + n_2 + 1 \leqslant i \leqslant n$ transmits

$$Y_i = \begin{cases} 1 & \text{if } X_i < \hat{\mu}_c, \\ 0 & \text{otherwise.} \end{cases}$$

---

[4]Here, one round of communication consists of bits sent from a subset of the users to the server, followed by a broadcast message sent by the server to the remaining users.

[5]If the protocol is sequential and user $i$ has access to $Y_1, \ldots, Y_{i-1}$, then $\hat{\mu}_c$ can be directly computed by the last $n_3$ users by observing the first $n_1 + n_2$ transmissions.

The server computes

$$F_3 \triangleq \frac{1}{n_3} \sum_{i=n_1+n_2+1}^{n} Y_i \xrightarrow{\text{a.s.} \,|\, \hat{\mu}_c} F_X \left( \frac{\hat{\mu}_c - \mu}{\sigma} \right),$$

where almost sure convergence follows from the strong law of large numbers, conditional on $\hat{\mu}_c$, as $n_3 \to \infty$. Let

$$\alpha_3 \triangleq F_X^{-1}(F_3).$$

The final estimate of $\mu$ is then

$$\hat{\mu}_f \triangleq \hat{\mu}_c - F_X^{-1}(F_3) \cdot \hat{\sigma}_c, \tag{16}$$

which converges almost surely to $\mu$ as $n \to \infty$ by the continuous mapping theorem.

**Theorem 4.1.** *Consider the adaptive protocol of Sec. 4 with $n_1 = n_2 = \frac{n_3}{\log n_3}$ and $n_1 + n_2 + n_3 = n$. Then, for all $\mu \in \mathbb{R}$ and $\sigma > 0$,*

$$\sqrt{n} \, (\hat{\mu}_f - \mu) \xrightarrow{d} \mathcal{N} \left( 0, \frac{\sigma^2}{4 f_X(0)^2} \right),$$

*and the asymptotic MSE satisfies*

$$\lim_{n \to \infty} n \cdot \text{MSE}(\hat{\mu}_f) = \frac{\sigma^2}{4 f_X(0)^2}. \tag{17}$$

See Appendix C for the proof.

*Remark* 4.2. It may seem that the above protocol is not making full use of the power of adaptive protocols, as the last $n_3$ users are not making use of each others' transmissions. Similar to the observation made in (Kipnis & Duchi, 2022) for location families, we can show that increasing the number of rounds does not give us a lower asymptotic mean squared error. Moreover, for a large subclass of distributions, we can derive a matching lower bound, thereby establishing the optimality of our simple two-round protocol.

### 4.2 Lower Bounds for Adaptive Protocols

The following theorem gives a lower bound for adaptive protocols and is proved in Appendix D.

**Theorem 4.3.** *Let $f_X$ be a symmetric log-concave distribution for which*

$$\eta(x) \triangleq \frac{f_X^2(x)}{F_X(x) F_X(-x)} \tag{18}$$

*is non-increasing in $|x|$ and is uniquely maximized at $x = 0$. Consider any prior density $g$ on $\mu$ such that $\mathbb{E}\left[ (g'(\mu)/g(\mu))^2 \right]$ is bounded, where $g'$ denotes the derivative of $g$. Then, any 1-bit adaptive protocol must satisfy*

$$\lim_{n \to \infty} n \cdot \mathbb{E}[(\hat{\mu} - \mu)^2] \geq \frac{\sigma^2}{4 f_X^2(0)}, \tag{19}$$

*for every $\sigma > 0$. The expectation is taken with respect to the joint density $g(\mu) f_{X,\mu,\sigma}(x)$ on $(\mu, \sigma)$.*

It should also be noted that the above result assumes that the protocol is sequential — where $Y_i$ is allowed to depend only on $X_i$ and $Y_1, \ldots, Y_{i-1}$ — and does not hold for more complicated (e.g., blackboard) protocols.

Our lower bound in the adaptive case (equation 19) holds for symmetric, log-concave distributions such as Gaussian, logistic, hyperbolic secant, and generalized Gaussian. Several distributions satisfy the assumption in Theorem 4.3, including generalized normal (generalized Gaussian) distributions with shape parameter $1 < \beta \leq 2$ (this includes, in particular, the Laplace distribution ($\beta = 1$ as a limiting case) and the normal distribution ($\beta = 2$)). Symmetric log-concave distributions that fail this assumption include the uniform distribution and the generalized normal distribution with shape parameter $\beta > 2$. Theorem 4.3 requires the shape condition (defined in equation 18) to be non-increasing in $|x|$ and uniquely maximized at $x = 0$. This ensures that the Fisher information from one-bit messages is maximized when the quantization threshold is placed at the true mean.

| $Z_{\mathrm{std}}$ | $f_X(0)$ | $x^\star$ | $h^\star$ | $T(f_X)$ |
|---|---|---|---|---|
| 0.8665 | 0.4246 | 0.4854 | 0.4607 | 0.0246 |

Table 1: Numerical evaluation of the normalizing constant $Z_{\mathrm{std}}$, the density at the origin $f_X(0)$, the maximizer $x^\star$ of $h(x)$, the corresponding value $h^\star$, and the distribution-dependent constant $T(f_X)$ for the example equation 20

| Distribution | $C_{\mathrm{non}}$ | $C_{\mathrm{adapt}}$ | $C_{\mathrm{non}}/C_{\mathrm{adapt}}$ |
|---|---|---|---|
| Generalized Gaussian ($\beta = 1.5$) | 2.5806 | 1.1035 | 2.3385 |
| Logistic | 1.1619 | 1.2159 | 0.9556 |
| Hyperbolic secant | 1.1239 | 1.0000 | 1.1239 |
| Sin2 (custom example equation 20) | 4.1982 | 1.3868 | 3.0272 |

Table 2: Adaptive vs. non-adaptive constants for unit-variance distributions. Generalized Gaussian uses $\beta = 1.5$. Values computed using $C_{\mathrm{non}} = 0.1034/T(f_X)$ and $C_{\mathrm{adapt}} = 1/(4f_X(0)^2)$.

### 4.3 Suboptimality of Non-Adaptive Protocols

For any symmetric strictly log-concave $f_X$ satisfying equation 18, the benchmark upper bound on the asymptotic value of $n\mathrm{MSE}(\hat\mu)/\sigma^2$ for adaptive protocols is

$$C_{\mathrm{adapt}} \triangleq \frac{1}{4f_X^2(0)},$$

and this can be achieved by our simple two-round protocol. The corresponding lower bound for non-adaptive protocols is

$$C_{\mathrm{non}} = \frac{0.1034}{T(f_X)}.$$

These correspond to the normalized (by $1/\sigma^2$) upper and lower bounds derived in Theorem 4.3 and Theorem 3.2 respectively. Although the term $T(f_X)$ does not have a simple closed-form expression, this depends only on $f_X$ and hence can be evaluated numerically.

We show that for many distributions of interest, $C_{\mathrm{adapt}} < C_{\mathrm{non}}$, implying that non-adaptive protocols are strictly suboptimal compared to adaptive protocols.

As a toy example, consider the following strictly log-concave distribution,

$$f_X(x) = \frac{1}{Z_{\mathrm{std}}} e^{-\psi(x)}. \tag{20}$$

where

$$\psi(x) = 1.48 \left(\frac{|x|}{2.023076}\right)^{1.5} + 0.5 \left(\frac{x}{2.023076}\right)^4 + 0.0675 \sin^2\left(4\,\frac{x}{2.023076}\right) + 1,$$

and

$$Z_{\mathrm{std}} = \int_{-\infty}^{\infty} \exp\left(-\left[1.48\left(\frac{|x|}{2.023076}\right)^{1.5} + 0.5\left(\frac{x}{2.023076}\right)^4 + 0.0675\sin^2\left(4\,\frac{x}{2.023076}\right) + 1\right]\right)\,dx.$$

Here, the constants are chosen so that $f_X$ is a valid density and has unit variance.

Substituting for $\phi'$ and $h'$, we numerically evaluate these integrals, and the results are listed in Table 1. For this distribution, we find that $C_{\mathrm{adapt}} = 1.3868$ whereas $C_{\mathrm{non}} = 4.1982$, thereby demonstrating that in this case the best non-adaptive protocol has strictly *larger* asymptotic MSE than adaptive protocols.

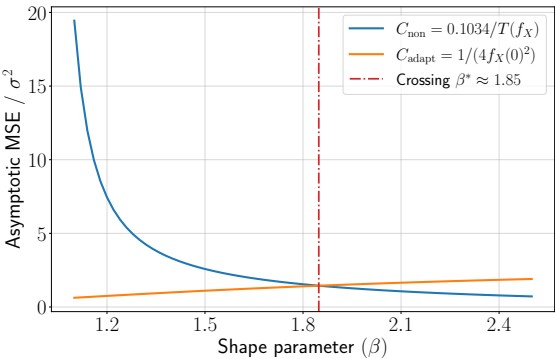

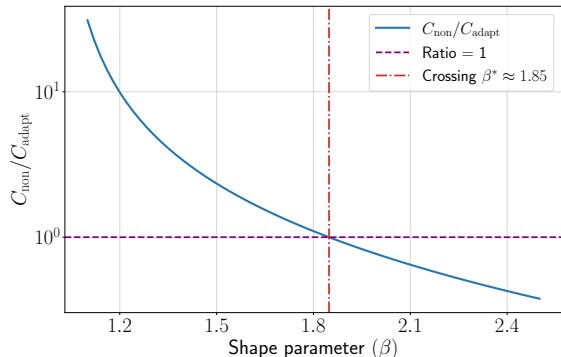

Figure 3: Illustration of bounds on $\frac{1}{\sigma^2}\lim_{n\to\infty}\mathrm{MSE}(\hat{\mu})$ for the generalized Gaussian family parameterized by $\beta > 1$. Here, $C_{\mathrm{non}}$ denotes the lower bound on this quantity for non-adaptive protocols, whereas $C_{\mathrm{adapt}}$ denotes the corresponding upper bound for our simple adaptive protocol. As $\beta$ increases, $C_{\mathrm{non}}$ decreases while $C_{\mathrm{adapt}}$ increases. The curves intersect at $\beta \approx 1.8488$. This allows us to conclude that non-adaptive protocols are *provably suboptimal* compared to adaptive protocols for $\beta < 1.8488$.

Figure 4: Ratio of constants $C_{\mathrm{non}}/C_{\mathrm{adapt}}$ for the unit-variance Generalized Gaussian Distribution (GGD) as a function of the shape parameter $\beta \in [1.1, 1.85]$. We compute $C_{\mathrm{non}} = \frac{0.1034}{T(f_X)}$ (non-adaptive lower-bound constant) and $C_{\mathrm{adapt}} = \frac{1}{4f_X(0)^2}$ (adaptive constant) from the unit-variance density $f_X$. The curves cross at $\beta^\star \approx 1.85$. For $\beta < \beta^\star$ the non-adaptive lower bound exceeds the adaptive constant (ratio > 1), while for $\beta > \beta^\star$ the adaptive constant is larger (ratio < 1).

For the Generalized Gaussian family of distributions (GGD)

$$f_X(x) = \frac{\beta}{2\alpha\Gamma(1/\beta)}\exp\left(-(|x|/\alpha)^\beta\right), \qquad \alpha \triangleq \sqrt{\frac{\Gamma(1/\beta)}{\Gamma(3/\beta)}}, \tag{21}$$

the density has zero mean and unit variance for all $\beta > 0$, and is strictly log-concave for $\beta > 1$. Note that $\beta = 2$ gives the standard normal. For this family, we find that the non-adaptive asymptotic constant is guaranteed to be larger than the adaptive constant throughout the range $1 < \beta < 1.85$. For lighter-tailed shapes with $\beta > 1.85$, this inequality no longer holds, and the adaptive constant becomes smaller. See Fig. 3 and Fig. 4. We conjecture that the lower bound is weak for $\beta > 1.85$ and there is scope for improvement.

Note that Theorem 3.2 explicitly requires strict log-concavity. The Laplace distribution is log-concave but not strictly log-concave. Therefore, the adaptive lower bound remains valid for this distribution, while the non-adaptive lower bound does not apply since it requires strict log-concavity.

## 4.4 Communication and computational complexity

Both the non-adaptive and adaptive protocols require only one bit of communication per user, yielding a total communication cost of $n$ bits. The adaptive protocol involves two rounds. In the first round, $n_1 + n_2$ users transmit one-bit messages based on local thresholding operations. The last $n_3$ users require access to the estimate $\hat{\mu}_c$ formed from the earlier transmissions. This can be implemented in several standard ways. Under a sequential model, the last $n_3$ users may view the previously posted one-bit messages and compute $\hat{\mu}_c$ directly. In settings without such shared access, the server may broadcast a single real number to these users, which incurs only a constant amount of additional communication (assuming a broadcast model of communication). The computational cost is also minimal. Each user performs an $O(1)$ thresholding step, and the server aggregates $O(n)$ one-bit messages and then performs a constant number of real-valued operations. Hence the overall computational complexity is $O(1)$ for each user and $O(n)$ for the server.

## 4.5 Increasing bit budgets provide marginal gains

While the main focus of our work is on one-bit compressors, it is helpful to compare our bounds on the asymptotic MSE with a corresponding lower bound under the *centralized setting*, or the case where *no*

*communication constraints* are imposed on the users. This helps assess whether allowing more than one bit per sample can yield a substantial reduction in MSE.

The Fisher information for the location in the location-scale family for a given $\sigma > 0$ is

$$I_F(f_X; \sigma) = \frac{1}{\sigma^2} \int_{\mathbb{R}} \frac{[f'_X(z)]^2}{f_X(z)} \, dz.$$

For any regular estimator based on $n$ i.i.d. samples, the Cramér–Rao bound implies that the MSE of any unbiased estimator is

$$\mathrm{MSE}(\hat{\mu}_n) \geqslant \frac{\sigma^2}{n \, J(f_X)}, \qquad J(f_X) = \int_{\mathbb{R}} \frac{[f'_X(z)]^2}{f_X(z)} \, dz.$$

Equivalently, the Van-Trees inequality (Gill & Levit, 1995) yields an asymptotic lower bound for any (potentially biased) estimator

$$\lim_{n\to\infty} n\mathrm{MSE}(\hat{\mu}_n) \geqslant \frac{\sigma^2}{J(f_X)},$$

where the expectation is taken with respect to $f_{X,\mu,\sigma} f_\theta$ for any prior density on $f_\theta$ for which the above exists.

Thus $1/J(f_X)$ can be thought of as the best $1/n$ constant achievable in the absence of quantization.

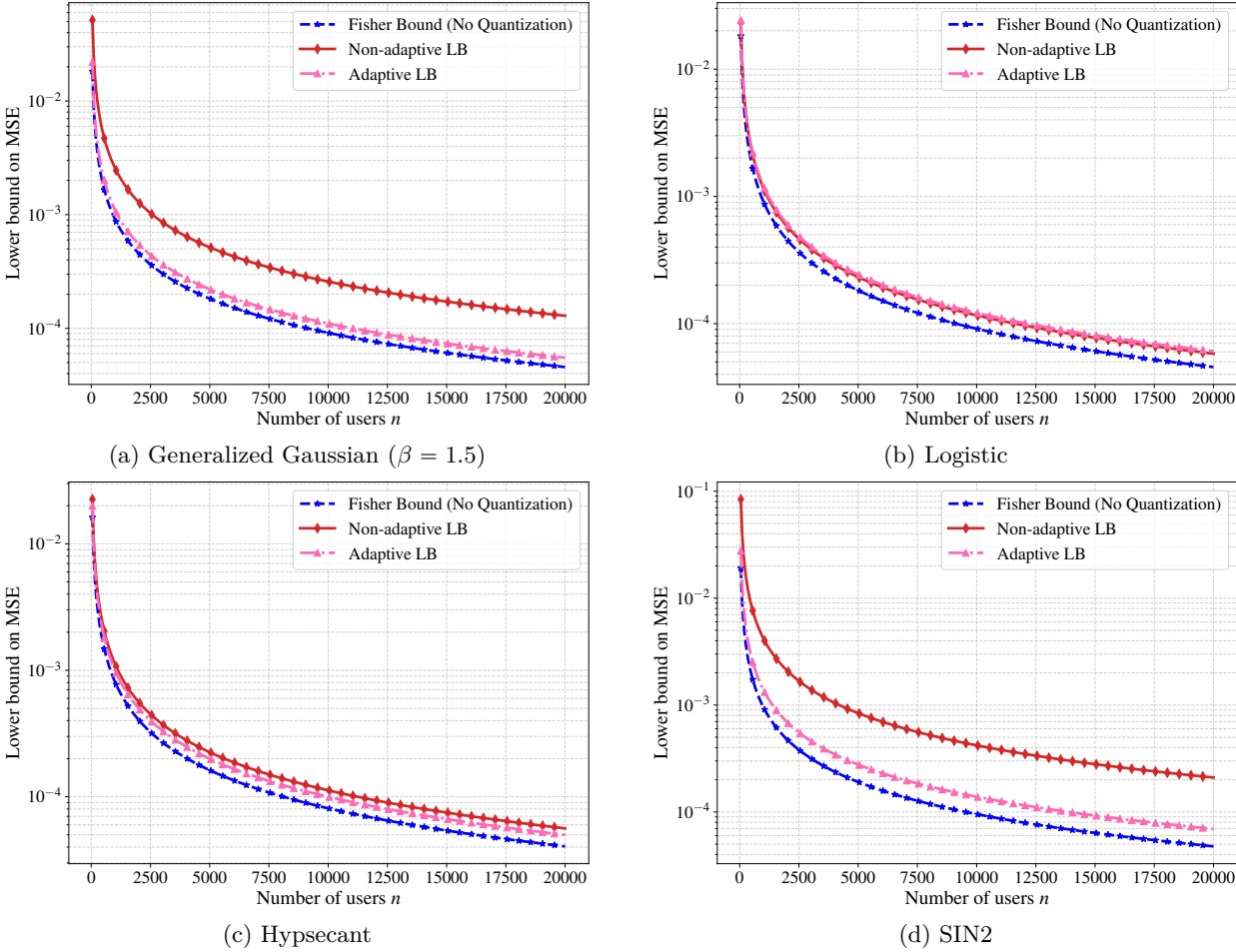

Figure 5: Fisher-information lower bound (no quantization) compared with the adaptive and non-adaptive one-bit lower bounds for four distributions. All curves decay as $1/n$; differences arise only through the proportionality constants. The adaptive lower bound is tight, which implies that increasing the number of bits per user can only provide a minor improvement in the asymptotic MSE.

| Distribution | $C_{\mathrm{FI}}$ | $C_{\mathrm{adp}}$ | $C_{\mathrm{non}}$ |
|---|---|---|---|
| Gaussian (GGD, $\beta = 1.5$) | 0.9126 | 1.1035 | 2.5806 |
| Logistic | 0.9119 | 1.2159 | 1.1619 |
| Hypersecant | 0.8106 | 1.0000 | 1.1239 |
| SIN2 | 0.9534 | 1.3868 | 4.1982 |

Table 3: Normalized constants $(n \, \mathrm{MSE})/\sigma^2$ for the Fisher-information bound, the adaptive one-bit bound, and the non-adaptive bound, evaluated using standardized densities with $\mu = 0$ and $\sigma = 1$.

Figure 5 illustrates the Fisher-information bound (centralized setting) together with the adaptive and non-adaptive one-bit lower bounds we derived. All bounds exhibit the same dependence on $n, \sigma$. The adaptive one-bit curve remains close to the centralized benchmark. We also report

$$C_{\mathrm{FI}} = \frac{1}{J(f_X)}, \qquad C_{\mathrm{adp}} = \frac{1}{4 f_X^2(0)}, \qquad C_{\mathrm{non}} = \frac{\alpha^*}{T(f_X)},$$

in Table 3.

The results exhibit a consistent pattern across all distributions. The adaptive one-bit method attains a constant close to the Fisher-information limit, which represents the best performance achievable without quantization. Consequently, the potential gain from transmitting more than one bit per sample is limited in this setting.

The behavior of the non-adaptive bound is determined by the functional $T(f_X)$, and we do not claim optimality of this bound. As seen in Table 3, the constant $C_{\mathrm{non}}$ may be either larger or smaller than the adaptive constant, depending on the underlying distribution. The comparison with the Fisher-information bound should therefore be understood as a consequence of the particular form of our non-adaptive bound, rather than as a fundamental limitation of non-adaptive schemes.

## 5 Simulation Results

We validate our theoretical bounds by empirically evaluating the worst-case and average (over $\mu$) MSE for four representative symmetric, strictly log-concave source distributions: generalized Gaussian with $\beta = 1.5$, Logistic, Hyperbolic secant, and the custom density in equation 20. Each experiment averages over $N_{\mathrm{trials}} = 2000$ trials. The unknown mean $\mu$ is varied over the interval $[-2.5, 2.5]$ in increments of 0.5, while the standard deviation is fixed at $\sigma = 2$.

Quantization thresholds $\theta_1$ and $\theta_2$ are precomputed from the known range $(\mu_{\min}, \mu_{\max})$ to ensure consistency across trials. Specifically, we adopt the equal-thirds rule:

$$\theta_1 = \mu_{\min} + \frac{1}{3}(\mu_{\max} - \mu_{\min}), \qquad \theta_2 = \mu_{\min} + \frac{2}{3}(\mu_{\max} - \mu_{\min}).$$

Among all possible choices of two thresholds, this rule minimizes the maximum distance from any $\mu \in [\mu_{\min}, \mu_{\max}]$ to its nearest threshold, as expressed by

$$\min_{\theta_1, \theta_2} \max_{\mu \in [\mu_{\min}, \mu_{\max}]} \min_{i \in \{1,2\}} |\mu - \theta_i|.$$

In addition, for symmetric strictly log-concave scale-location densities, placing thresholds symmetrically around the mid-range $(\mu_{\min} + \mu_{\max})/2$ prevents saturation of the empirical c.d.f. estimates $F_1, F_2$ and keeps the Bernoulli variances $F_j(1 - F_j)$ bounded away from zero. This yields stable estimation of both $\mu$ and $\sigma$, while remaining distribution-agnostic. Finally, precomputing thresholds from the range guarantees fairness across trials and enforces strict non-adaptivity, which is essential when contrasting against adaptive protocols. Our implementation code is publicly available on GitHub and can be accessed at Kumar & Vatedka (2026b)

The simulation curves in Fig. 6 demonstrate the superiority of adaptive protocols over non-adaptive protocols. Although the lower bounds that we derive are asymptotic and are not actually valid for small $n$, we

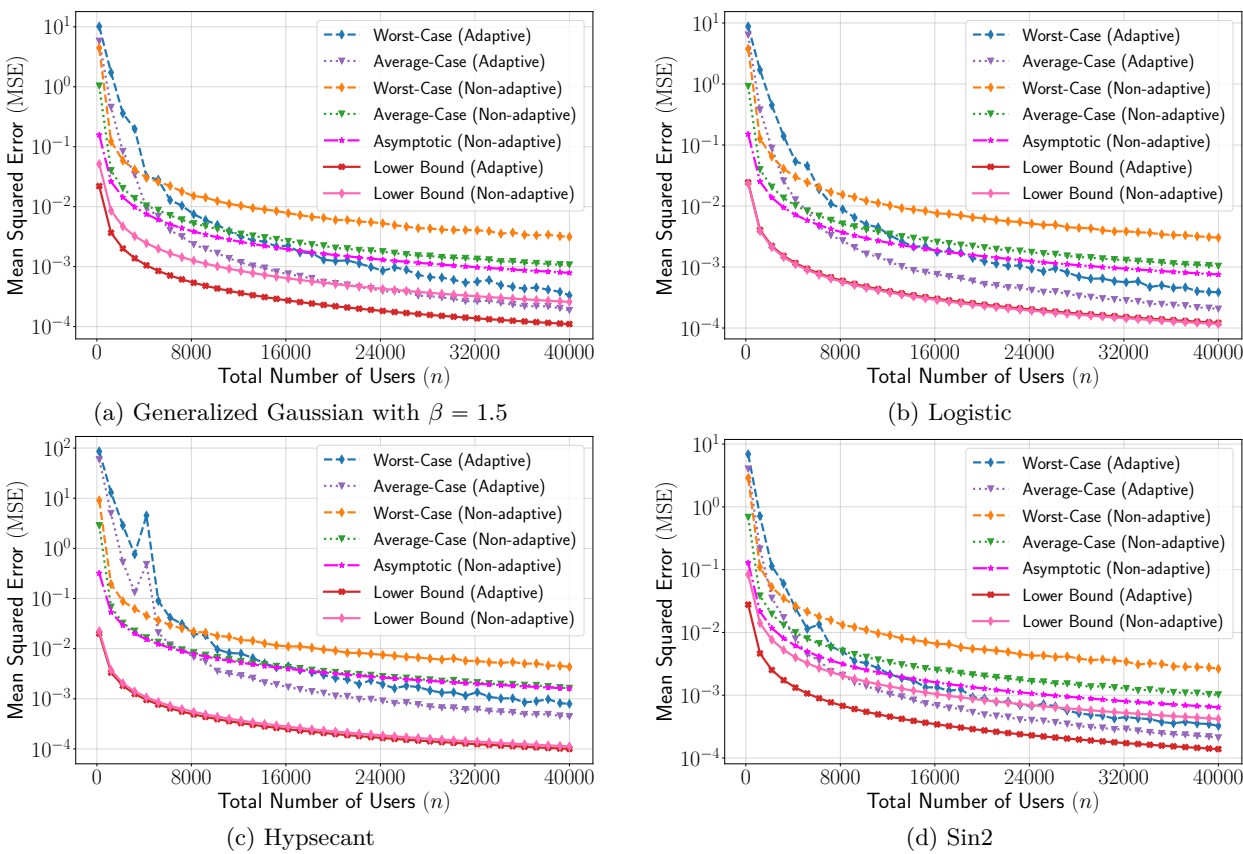

Figure 6: Worst-case and average (over $\mu$) MSE across four source distributions under one-bit protocols. Benchmarks are computed using $f_X(0)$ for each family; adaptive and non-adaptive curves are shown according to the legend. The curve labeled *Asymptotic (Non-adaptive)* represents the asymptotic value of MSE predicted by Theorem 3.1.

nevertheless plot these and can be interpreted as the first-order approximations of the lower bounds. We also observe that in the case of the generalized Gaussian, the plot for empirical worst-case MSE intersects the lower bound for the asymptotic MSE for non-adaptive estimators. It should be noted that the lower bound for non-adaptive estimators could potentially be loose, which illustrates that our simple adaptive estimator beats the best non-adaptive estimator even for finite $n$.

In the non-adaptive protocol, the total number of users $n$ is split into two disjoint groups $n_1 = K_1 n$ and $n_2 = K_2 n$ with $K_1 + K_2 = 1$. Since the protocol remains consistent for any fixed $(K_1, K_2)$, we explored a range of allocations to study the effect of the split on finite-sample performance. In particular, we considered $(K_1, K_2) \in \{(0.10, 0.90), (0.20, 0.80), (0.30, 0.70), (0.40, 0.60), (0.50, 0.50)\}$. The resulting curves show that different splits slightly shift the MSE at moderate sample sizes, but the overall asymptotic behavior is unchanged, and in all cases the non-adaptive MSE remains strictly above the adaptive benchmark. This confirms that the suboptimality of non-adaptive protocols is inherent and not due to a particular choice of partition.

In the adaptive protocol, users are divided into three groups of sizes $n_1, n_2, n_3$ with $n_1 + n_2 + n_3 = n$. The first two groups provide coarse estimates $(\hat{\mu}_c, \hat{\sigma}_c)$, which are then refined using the remaining $n_3$ users. To check robustness, we considered small fixed first-stage allocations with $(K_1, K_2) \in \{(0.05, 0.05), (0.10, 0.10), (0.15, 0.15)\}$, showing that even a small initial budget is sufficient for effective refinement. We also evaluated the theoretically motivated split $n_1 = n_2 = n_3 / \log n_3$ from Theorem 4.1, which achieves the asymptotic MSE. Together, these experiments suggest that the adaptive scheme is both practical (performing well with small first-stage budgets) and information-theoretically optimal.

We also compute asymptotic constants $C_{non}$ and $C_{adapt}$ for the four families, and are tabulated in Table 2.

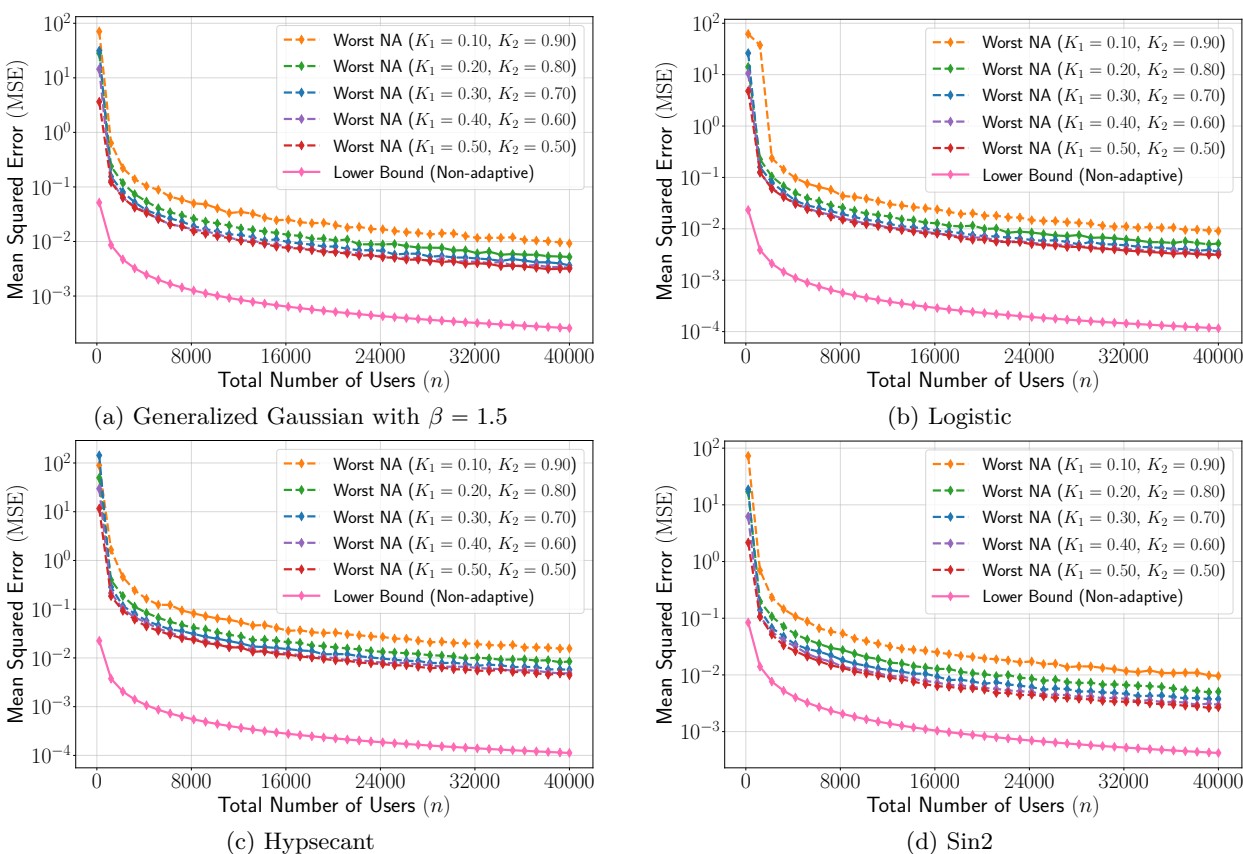

Figure 7: Non-adaptive MSE for symmetric, strictly log-concave distributions with different $K_1, K_2$ allocations. Each subfigure shows worst-case MSE alongside the lower bound $\frac{0.1043}{T(f_X)}\frac{\sigma^2}{n}$. A consistent positive gap is observed across all cases.

Finally, Figs. 7 and 8 analyze the effect of varying $(n_1, n_2)$. The eight combined plots show worst-case MSE on a log scale across all four distributions. In the non-adaptive case, we fix $(\theta_1, \theta_2)$ as described earlier but vary $K_1 = n_1/n$ and $K_2 = n_2/n$.

Fig. 9 illustrates the performance of the allocation strategy from Theorem 4.1, where $n_1 = n_2 = n_3/\log n_3$. Here, both worst-case and average MSE curves gradually approach the lower bound. This confirms that the theorem-based allocation achieves optimality, and that even with alternative user splits, the asymptotic behavior of the adaptive scheme remains essentially unchanged.

For the generalized Gaussian source with $\beta = 1.50$ and unit variance, Fig. 10 illustrates the worst-case MSE as $\mu \in [-2.5, 2.5]$ with $n = 40000$. Thresholds are fixed once using the equal-thirds rule ($\theta_1 = -0.8333$, $\theta_2 = 0.8333$). The MSE achieved by the adaptive protocol is lower the non-adaptive lower bound for large enough $n$, providing a clear empirical validation of our results.

We also examine the standard normal, which corresponds to the special case of the generalized Gaussian distribution with $\beta = 2$. As shown in Fig. 11, using the same experimental setup as Fig. 6, the non-adaptive lower bound is observed to lie below the adaptive bound. However, we conjecture that the lower bound for non-adaptive protocols can be improved.

*Remark* 5.1. These experiments also show how the asymptotic behavior appears in the finite-user setting. Our results identify the leading term

$$\text{MSE}(n) = \frac{C}{n} + o(1/n),$$

with $C$ determined by the choice of protocol and the underlying source distribution. Although this expression does not capture higher-order corrections, the simulations indicate that the $C/n$ approximation

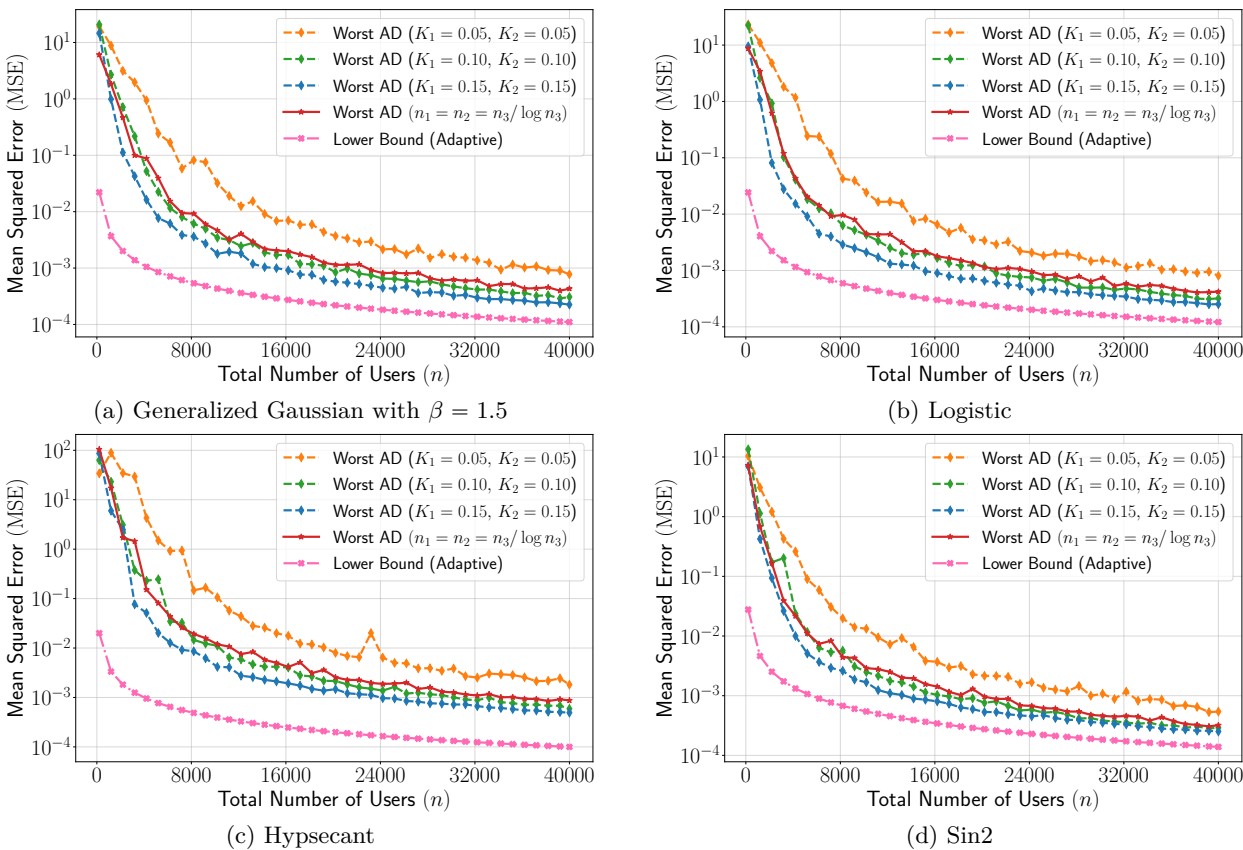

Figure 8: Adaptive MSE across four symmetric, strictly log-concave distributions. Worst-case closely follow the benchmark $\frac{\sigma^2}{4f_X(0)^2 n}$, confirming asymptotic optimality.

remains accurate for moderately large values of $n$. Across all distributions considered, the empirical curves follow the $1/n$ behavior, indicating that the asymptotic constants closely describe the observed finite-sample performance.

# 6 Conclusion and Future Work

In this work, we derived the asymptotic mean squared error (MSE) for estimating the location parameter of distributions from the scale-location family under one-bit communication constraints, with a primary focus on the case where the variance is unknown. Our main contribution is the design and analysis of both non-adaptive and adaptive protocols for mean estimation. As a byproduct, we also proposed a simple non-adaptive protocol for the standard deviation and derived its asymptotic MSE. We note, however, that our results for variance estimation apply only to the non-adaptive setting, and we do not claim optimality for this estimator. We also derived a non-parametric 1-bit estimator for the mean, but observed that the MSE scales only as $\omega(1/n)$, which is worse than our simple non-adaptive estimator.

For a broad class of symmetric strictly log-concave distributions, we proved that no one-bit adaptive protocol can achieve an asymptotic MSE smaller than presented in Theorem 4.3, and we designed an adaptive scheme that attains this bound, thereby establishing its information-theoretic optimality. In contrast, we showed that one-bit non-adaptive protocols necessarily incur a strictly larger MSE for many such distributions, and we quantified the exact performance gap between optimal non-adaptive and adaptive schemes in terms of a distribution-dependent constant $T(f_X)$.

Our numerical comparison with the unquantized Fisher-information bound shows that the adaptive one-bit scheme already operates close to the best performance achievable without communication constraints. This

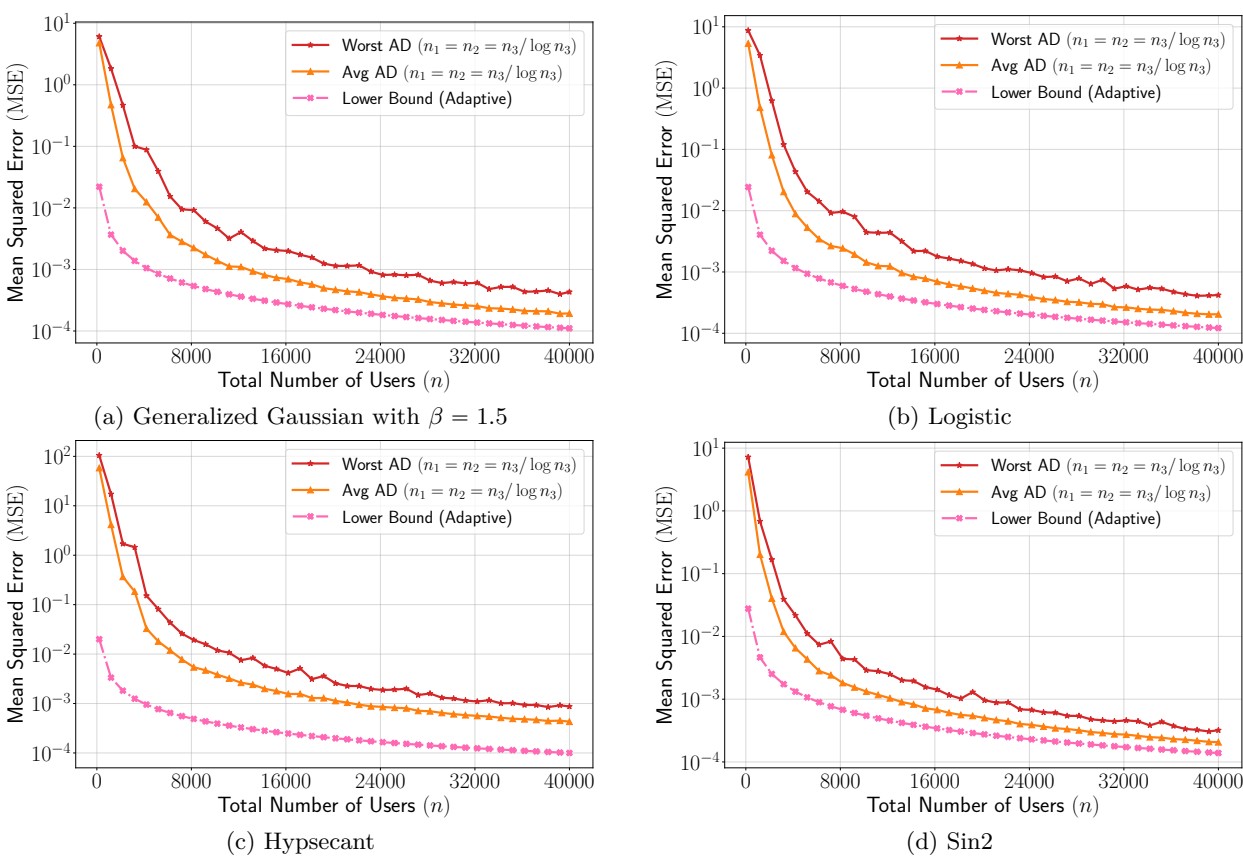

Figure 9: Adaptive MSE across four symmetric, strictly log-concave distributions with the special split $n_1 = n_2 = n_3/\log n_3$ used in Theorem 4.1. Both worst-case and average MSE approach the benchmark $\frac{\sigma^2}{4f_X(0)^2 n}$ asymptotically.

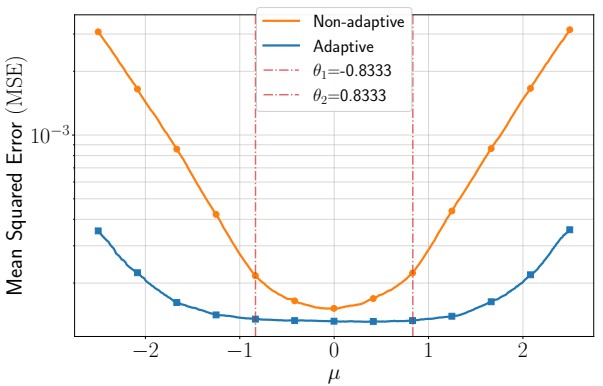

Figure 10: MSE versus the mean $\mu$ for the generalized Gaussian source with shape parameter $\beta = 1.50$ and unit variance, using $n = 40000$ users. Thresholds are fixed at $\theta_1 = -0.8333$ and $\theta_2 = 0.8333$, and $\mu$ is varied over $[-2.5, 2.5]$.

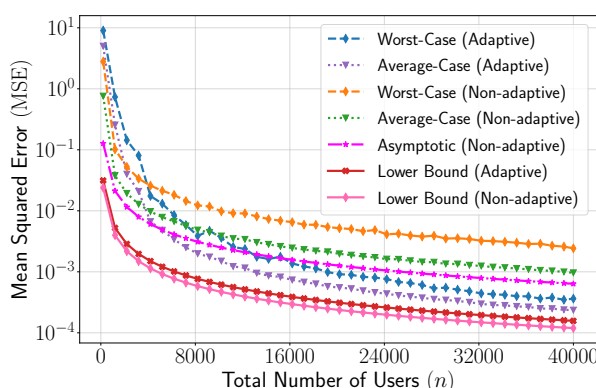

Figure 11: Worst-case and average MSE for the standard normal (GGD with $\beta = 2$) under one-bit protocols. The *Asymptotic (Non-adaptive)* curve is the asymptotic MSE predicted by Theorem 3.1. Although the lower bound for non-adaptive protocols is lower than that for adaptive protocols, we conjecture that the lower bound is in fact loose and can be improved.

suggests that, within the strictly log-concave family, the improvement in MSE obtained by transmitting more than one bit per sample is limited.

We believe that the lower bound for non-adaptive protocols is not tight, and can be improved. We conjecture that for all symmetric log-concave distributions $f_X(x) = e^{-\phi(x)}$ where $\phi$ is upper bounded by some fixed polynomial, the asymptotic minimax for non-adaptive estimators is strictly higher than that achieved by our adaptive estimator.

While our analysis addresses the scalar mean estimation problem, several extensions remain open. In particular, extending the lower bound techniques to high-dimensional vectors, exploring protocols under multi-bit communication budgets (e.g., $b$ bits per sample), and developing adaptive strategies for optimal variance (or scale) estimation are promising directions for future work. Our techniques could potentially also be used to derive upper and lower bounds for estimation of higher-order moments or more general exponential families. Even in the context of mean estimation, deriving tight bounds on the minimax MSE for general log-concave distributions that hold for finite $n$ remains an open question.

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

## A  Proof of Theorem 3.1

We use the delta method (Casella & Berger, 2002, Theorem 5.5.24) and standard limit theorems to prove this result. Using the central limit theorem (Casella & Berger, 2002, Theorem 5.5.14), we can write:

$$\sqrt{n_1}\left(F_1 - F_X\left(\frac{\theta_1 - \mu}{\sigma}\right)\right) \xrightarrow{d} \mathcal{N}\left(0, \sigma_1^2\right),$$

$$\sqrt{n_2}\left(F_2 - F_X\left(\frac{\theta_2 - \mu}{\sigma}\right)\right) \xrightarrow{d} \mathcal{N}\left(0, \sigma_2^2\right).$$

Using the delta method (Casella & Berger, 2002, Theorem 5.5.24), we can write:

$$\sqrt{n_1}\left(\alpha_1 - \left(\frac{\theta_1 - \mu}{\sigma}\right)\right) \xrightarrow{d} \left(\frac{dF_X^{-1}(z)}{dz}\bigg|_{z=F_X\left(\frac{\theta_1-\mu}{\sigma}\right)}\right)\omega_1,$$

$$\sqrt{n_2}\left(\alpha_1 - \left(\frac{\theta_2 - \mu}{\sigma}\right)\right) \xrightarrow{d} \left(\frac{dF_X^{-1}(z)}{dz}\bigg|_{z=F_X\left(\frac{\theta_2-\mu}{\sigma}\right)}\right)\omega_2,$$

$$(22)$$

where $\omega_i \sim \mathcal{N}(0, \sigma_i^2)$ for $i = 1, 2$. After solving equation 22 we get:

$$\sqrt{n_1}\left(\alpha_1 - \left(\frac{\theta_1 - \mu}{\sigma}\right)\right) \xrightarrow{d} \mathcal{N}\left(0, \frac{\sigma_1^2}{f_X^2(\frac{\theta_1-\mu}{\sigma})}\right),$$

$$\sqrt{n_2}\left(\alpha_1 - \left(\frac{\theta_2 - \mu}{\sigma}\right)\right) \xrightarrow{d} \mathcal{N}\left(0, \frac{\sigma_2^2}{f_X^2(\frac{\theta_2-\mu}{\sigma})}\right).$$

$$(23)$$

Let us define a function $\xi$ as:

$$\xi(\alpha_1, \alpha_2) \triangleq \frac{\alpha_1\theta_2 - \alpha_2\theta_1}{\alpha_1 - \alpha_2} \tag{24}$$

Using the delta method (Casella & Berger, 2002, Theorem 5.5.24), we can write:

$$\sqrt{n}\left(\xi(\alpha_1, \alpha_2) - \xi\left(\frac{\theta_1 - \mu}{\sigma}, \frac{\theta_2 - \mu}{\sigma}\right)\right) \xrightarrow{d} \mathcal{N}\left(0, \nabla\xi^T\Sigma\nabla\xi\right)$$

where $n = n_1 + n_2$. Now we have

$$\nabla\xi\bigg|_{\substack{\alpha_1 = \frac{\theta_1-\mu}{\sigma}, \\ \alpha_2 = \frac{\theta_2-\mu}{\sigma}}} = \left(\frac{\sigma(\theta_2 - \mu)}{\theta_1 - \theta_2}, \frac{\sigma(\theta_1 - \mu)}{\theta_1 - \theta_2}\right)^T,$$

and

$$\Sigma = \begin{pmatrix} \frac{K_1 \sigma_1^2}{f_x^2(\frac{\theta_1 - \mu}{\sigma})} & 0 \\ 0 & \frac{K_2 \sigma_2^2}{f_x^2(\frac{\theta_2 - \mu}{\sigma})} \end{pmatrix}.$$

Recall that $n_1 = K_1 n$ and $n_2 = K_2 n$, where $K_1$ and $K_2$ are constants independent of $n$, and $K_1 + K_2 = 1$. Now

$$\nabla \xi^T \Sigma \nabla \xi = \frac{\sigma^2}{(\theta_1 - \theta_2)^2} \left[ (\theta_2 - \mu)^2 \frac{K_1 \sigma_1^2}{f_x^2(\frac{\theta_1 - \mu}{\sigma})} 2 + (\theta_1 - \mu)^2 \frac{K_2 \sigma_2^2}{f_x^2(\frac{\theta_2 - \mu}{\sigma})} \right].$$

Since we have

$$\xi \left( \frac{\theta_1 - \mu}{\sigma}, \frac{\theta_2 - \mu}{\sigma} \right) = \mu,$$

we can write

$$\sqrt{n}(\mu - \hat{\mu}_c) \xrightarrow{d} \mathcal{N} \left( 0, \frac{\sigma^2}{(\theta_1 - \theta_2)^2} \left[ (\theta_2 - \mu)^2 \frac{K_1 \sigma_1^2}{f_x^2(\frac{\theta_1 - \mu}{\sigma})} 2 + (\theta_1 - \mu)^2 \frac{K_2 \sigma_2^2}{f_x^2(\frac{\theta_2 - \mu}{\sigma})} \right] \right)$$

Therefore, scheme will attain an asymptotic Mean Squared Error (MSE) of:

$$\lim_{n \to \infty} n \cdot \text{MSE}(\hat{\mu}_c) = \frac{\sigma^2}{(\theta_1 - \theta_2)^2} \left[ (\theta_2 - \mu)^2 \cdot \frac{K_1 \sigma_1^2}{f_x^2(\frac{\theta_1 - \mu}{\sigma})} + (\theta_1 - \mu)^2 \cdot \frac{K_2 \sigma_2^2}{f_x^2(\frac{\theta_2 - \mu}{\sigma})} \right]$$

This concludes the proof of the first part.

We now prove the second part. Using the continuous mapping theorem (Casella & Berger, 2002) in equation 23, we can write

$$\sqrt{n} \left( (\alpha_1 - \alpha_2) - \left( \frac{\theta_1 - \theta_2}{\sigma} \right) \right) \xrightarrow{d} \mathcal{N} \left( 0, \frac{K_1 \sigma_1^2}{f_X^2(\frac{\theta_1 - \mu}{\sigma})} + \frac{K_2 \sigma_2^2}{f_X^2(\frac{\theta_2 - \mu}{\sigma})} \right). \tag{25}$$

Now we define a function

$$\psi(x) \triangleq \frac{\theta_1 - \theta_2}{x}.$$

Then, its derivative

$$\psi' \left( \frac{\theta_1 - \theta_2}{\sigma} \right) = \frac{-\sigma^2}{\theta_1 - \theta_2}.$$

Applying the delta method (Casella & Berger, 2002), we can write

$$\sqrt{n} \left( \psi(\alpha_1 - \alpha_2) - \psi \left( \frac{\theta_1 - \theta_2}{\sigma} \right) \right) \xrightarrow{d} \mathcal{N} \left( 0, \left( \frac{\sigma^4}{(\theta_1 - \theta_2)^2} \right) \cdot \left( \frac{K_1 \sigma_1^2}{f_X^2(\frac{\theta_1 - \mu}{\sigma})} + \frac{K_2 \sigma_2^2}{f_X^2(\frac{\theta_2 - \mu}{\sigma})} \right) \right). \tag{26}$$

This gives us

$$\sqrt{n}(\sigma - \hat{\sigma}_c) \xrightarrow{d} \mathcal{N} \left( 0, \left( \frac{\sigma^4}{(\theta_1 - \theta_2)^2} \right) \left( \frac{K_1 \sigma_1^2}{f_X^2(\frac{\theta_1 - \mu}{\sigma})} + \frac{K_2 \sigma_2^2}{f_X^2(\frac{\theta_2 - \mu}{\sigma})} \right) \right).$$

Therefore

$$\lim_{n \to \infty} n \cdot \text{MSE}(\hat{\sigma}_c) = \left( \frac{\sigma^4}{(\theta_1 - \theta_2)^2} \right) \left( \frac{K_1 \sigma_1^2}{f_X^2(\frac{\theta_1 - \mu}{\sigma})} + \frac{K_2 \sigma_2^2}{f_X^2(\frac{\theta_2 - \mu}{\sigma})} \right).$$

This concludes the proof of the second part of the theorem. $\square$

# B    Upper Bound on Squared Hellinger Distance

**Theorem B.1.** *Let $f_X(x) = e^{-\phi(x)}$ be a symmetric log-concave distribution, with $\phi$ being strictly convex, differentiable, and upper bounded by a polynomial[6]. Let $f_{X,\mu_+,\sigma}$ and $f_{X,\mu_-,\sigma}$ be the distributions corresponding to the two hypotheses, where $\mu_+, \mu_-$ are defined in equation 11.*

*Let $\pi_i$ be any randomized function from $\mathbb{R}$ to $\{0, 1\}$ that is chosen independently of $X_i$, and $P_{Y_i}^+, P_{Y_i}^-$ be the distributions corresponding to $Y_i = \pi_i(X_i)$ when $X_i$ is drawn according to $f_{X,\mu_+,\sigma}$ and $f_{X,\mu_-,\sigma}$ respectively.*

*Then,*

$$H^2(P_{Y_i}^+, P_{Y_i}^-) \leqslant \epsilon^2 \left[ \int_0^{h^*} \phi'\left(h^{-1}(t)\right) h^{-1}(t) \, \mathrm{d}t + o(1) \right],$$

*where $o(1) \to 0$ as $\epsilon \to 0$. Here, $h(x) \triangleq 2\phi'(x) f_X(x)$ and $h^* = \max_{x \geqslant 0} h(x)$.*

To prove this theorem, we first introduce auxiliary functions that will be used in the analysis. Our construction partly follows (Cai & Wei, 2022a).

We define the maximum pointwise density difference

$$a_\epsilon^* \triangleq \sup_{x \in \mathbb{R}} |f_X(x + \epsilon) - f_X(x - \epsilon)|. \tag{27}$$

For $0 \leqslant a < a_\epsilon^*$, let

$$A_\epsilon(a) \triangleq \{x \in \mathbb{R} : |f_X(x + \epsilon) - f_X(x - \epsilon)| \geqslant a\}, \tag{28}$$

and

$$x_a(\epsilon) \triangleq \sup A_\epsilon(a). \tag{29}$$

Let

$$g_+(x, a) = \begin{cases} \frac{f_X(\epsilon)}{a_\epsilon^*}, & x = 0, \\ \frac{f_X(x+\epsilon)}{|f_X(x+\epsilon) - f_X(x-\epsilon)|}, & x \in A_\epsilon(a), \\ 0, & \text{otherwise}, \end{cases}$$

$$g_-(x, a) = \begin{cases} \frac{f_X(\epsilon)}{a_\epsilon^*}, & x = 0, \\ \frac{f_X(x-\epsilon)}{|f_X(x+\epsilon) - f_X(x-\epsilon)|}, & x \in A_\epsilon(a), \\ 0, & \text{otherwise}. \end{cases} \tag{30}$$

A direct calculation shows

$$\int_0^{a_\epsilon^*} g_+(x, a) \, \mathrm{d}a = f_X(x + \epsilon), \qquad \int_0^{a_\epsilon^*} g_-(x, a) \, \mathrm{d}a = f_X(x - \epsilon).$$

Thus,

$$\frac{1}{\sigma} f_X\left(\frac{x - \mu_-}{\sigma}\right) = \int_0^{a_\epsilon^*} \frac{1}{\sigma} g_+\left(\frac{x - \mu}{\sigma}, a\right) \, \mathrm{d}a,$$

and

$$\frac{1}{\sigma} f_X\left(\frac{x - \mu_+}{\sigma}\right) = \int_0^{a_\epsilon^*} \frac{1}{\sigma} g_-\left(\frac{x - \mu}{\sigma}, a\right) \, \mathrm{d}a.$$

Define the likelihood ratio

$$R_a \triangleq \sup_{x \in A_\epsilon(a)} \frac{f_X(x + \epsilon)}{f_X(x - \epsilon)}. \tag{31}$$

The next results show how $R_a$ can be used to bound the squared Hellinger distance.

---

[6]The exact value of the degree is not particularly important here. We only require the degree to be finite.

**Lemma B.2.** *Let $\pi_i : \mathbb{R} \to \{0, 1\}$ be any (possibly randomized) non-adaptive encoder, and let $g_+(\cdot, a), g_-(\cdot, a)$ be as in equation 30. Then, for $a \in [0, a_\epsilon^*)$ and any $\mu \in \mathbb{R}$, $\sigma > 0$, the induced distributions $P_{Y_i}^+$ and $P_{Y_i}^-$ satisfy*

$$H^2(P_{Y_i}^+; P_{Y_i}^-) \leqslant \int_0^{a_\epsilon^*} \frac{\sqrt{R_a} - 1}{\sqrt{R_a} + 1} \, x_a(\epsilon) \, \mathrm{d}a. \tag{32}$$

The proof can be found in Appendix G.

We split the integral above into two parts, and then further upper bound this using the following lemma.

**Lemma B.3.** *Let $f_X(x) = e^{-\phi(x)}$ where $\phi$ is even, differentiable, and strictly convex. For $\epsilon > 0$,*

$$\frac{\sqrt{R_a} - 1}{\sqrt{R_a} + 1} \leqslant \frac{\delta(x_a(\epsilon), \epsilon)}{4} \leqslant \frac{\epsilon \, \phi'(x_a(\epsilon) + \epsilon)}{2}, \tag{33}$$

*where*

$$\delta(x_a(\epsilon), \epsilon) \triangleq \phi(x + \epsilon) - \phi(x - \epsilon).$$

See Appendix G for the details (G.2).

Splitting equation 32 into two regions and then applying the two bounds from Lemma B.3, we have,

$$H^2(P_{Y_i}^+, P_{Y_i}^-) \leqslant \frac{1}{4} \int_0^{\epsilon^{2.5}} \delta(x_a(\epsilon), \epsilon) \, x_a(\epsilon) \, \mathrm{d}a + \frac{\epsilon}{2} \int_{\epsilon^{2.5}}^{a_\epsilon^*} \phi'(x_a(\epsilon) + \epsilon) \, x_a(\epsilon) \, \mathrm{d}a.$$
$$= I_1 + I_2 \tag{34}$$

- **First region** $(0 \leqslant a \leqslant \epsilon^{2.5})$: Here the integrand is small; a Taylor expansion near $a = 0$ shows the contribution is $o(\epsilon^2)$, negligible in the asymptotics.

- **Second region** $(\epsilon^{2.5} \leqslant a \leqslant a_\epsilon^*)$: This term is $\Theta(\epsilon^2)$, and we compute the underlying constant, which eventually yields the upper bound on MSE.

**First Region:** $0 \leqslant a \leqslant \epsilon^{2.5}$

Define

$$I_1 := \frac{1}{4} \int_0^{\epsilon^{2.5}} \delta(x_a(\epsilon), \epsilon) \, x_a(\epsilon) \, \mathrm{d}a.$$

Note that

$$a \leqslant f_X(x_a(\epsilon) - \epsilon) - f_X(x_a(\epsilon) + \epsilon) \leqslant f_X(x_a(\epsilon) - \epsilon)$$

and since $f_X(x) = e^{-\phi(x)}$, we have

$$x_a(\epsilon) \leqslant \phi^{-1}\left(\log \frac{1}{a}\right) + \epsilon,$$

where $\phi^{-1} : \mathbb{R}_{\geqslant 0} \to \mathbb{R}$ is guaranteed to exist as $\phi$ restricted to $\mathbb{R}_{\geqslant 0}$ is strictly convex, smooth and increasing; hence invertible. Under these assumptions on $\phi$, the inverse $\phi^{-1}$ is concave and smooth; see, e.g., (Niculescu & Persson, 2006, Prop. 1.1.7). Using first-order property of concave functions,

$$x_a(\epsilon) \leqslant \phi^{-1}(0) + \bar{\phi}(0) \log \frac{1}{a}, \tag{35}$$

where $\bar{\phi}(0)$ is the derivative of $\phi^{-1}$ evaluated at 0. Hence, $x_a(\epsilon)$ behaves as $O(\log(1/a))$ in this regime.

Let us now bound the integrand. Suppose that $\phi(x)$ is upper bounded by a polynomial of degree $k$, for some fixed $k$ (as assumed in the theorem). We have,

$$
\begin{aligned}
\delta(x_a(\epsilon), \epsilon) x_a(\epsilon) &= (\phi(x_a(\epsilon) + \epsilon) - \phi(x_a(\epsilon) - \epsilon)) x_a(\epsilon) \\
&\leqslant \max \left\{ \phi(x_a(\epsilon) + \epsilon), \phi(x_a(\epsilon) - \epsilon) \right\} x_a(\epsilon) \\
&= \phi(x_a(\epsilon) + \epsilon) x_a(\epsilon) \\
&\leqslant c \log^k \frac{1}{a}
\end{aligned}
\tag{36}
$$

for some universal constant $c$, as long as $a \leqslant \epsilon^{2.5}$. In the last step, we made use of equation 35 and the fact that $a \leqslant \epsilon^{2.5}$ and that $\phi$ is upper bounded by a polynomial of degree $k$. In this regime, $\log \frac{1}{a} = O(\log(1/\epsilon)) \to \infty$ as $\epsilon \to 0$, which is the dominant term.

Therefore, for some universal constants $c', c''$,

$$
\begin{aligned}
I_1 &\leqslant c' \int_0^{\epsilon^{2.5}} \log^k(a) \mathrm{d}a \\
&\leqslant c'' \epsilon^{2.5} \mathrm{poly} \left( \log \frac{1}{\epsilon} \right) \\
&\leqslant o(\epsilon^2)
\end{aligned}
\tag{37}
$$

where in the penultimate step, we have used the fact that the integral of $\log^k x$ is $x^2 \mathrm{poly}(\log x)$.

**Second Region:** $\epsilon^{2.5} \leqslant a \leqslant a_\epsilon^*$

Now

$$
I_2 := \frac{\epsilon}{2} \int_{\epsilon^{2.5}}^{a_\epsilon^*} \phi'(x_a(\epsilon) + \epsilon) \cdot x_a(\epsilon) \, \mathrm{d}a.
$$

From Taylors approximation, we have

$$
f_X(x - \epsilon) - f_X(x + \epsilon) = 2\epsilon f_X(x) \phi'(x) + O(\epsilon^3).
$$

Therefore,

$$
a_\epsilon^* \leqslant 2\epsilon \sup_{x \geqslant 0} f_X(x) \, \phi'(x) + O(\epsilon^3).
$$

Define

$$
h(x) \triangleq 2\phi'(x) f_X(x),
$$

and

$$
h^* \triangleq \sup_{x \geqslant 0} h(x).
$$

Then, for $a \in [\epsilon^{2.5}, a_\epsilon^*]$,

$$
x_a(\epsilon) = h^{-1} \left( \tfrac{a}{2\epsilon} + O(\epsilon^2) \right)
$$

Using this representation, we can bound $I_2$ as

$$
I_2 \leqslant \frac{\epsilon}{2} \int_{\epsilon^{2.5}}^{2\epsilon h^*} \phi' \left( h^{-1} \left( \tfrac{a}{2\epsilon} + O(\epsilon^2) \right) + \epsilon \right) \, h^{-1} \left( \tfrac{a}{2\epsilon} + O(\epsilon^2) \right) \, \mathrm{d}a.
$$

Applying the change of variable $t = \frac{a}{2\epsilon}$ gives

$$
I_2 \leqslant \epsilon^2 \int_{\epsilon^{1.5}/2}^{h^*} \phi' \left( h^{-1}(t + O(\epsilon^2)) + \epsilon \right) \, h^{-1}(t + O(\epsilon^2)) \, \mathrm{d}t.
$$

As $\epsilon \to 0$,

$$
I_2 \leqslant \epsilon^2 \left[ \int_{\epsilon^{1.5}/2}^{h^*} \phi' \left( h^{-1}(t) \right) \, h^{-1}(t) \, \mathrm{d}t + o(1) \right].
\tag{38}
$$

**Combined Bound**

Combining equation 37 and equation 38,

$$H^2(P_{Y_i}^+, P_{Y_i}^-) \leqslant o(\epsilon^2) + \epsilon^2 \left[ \int_{\epsilon^{1.5}/2}^{h^*} \phi'\left(h^{-1}(t)\right) h^{-1}(t)\,\mathrm{d}t + o(1) \right].$$

and hence,

$$H^2(P_{Y_i}^+, P_{Y_i}^-) \leqslant \epsilon^2 \left[ \int_0^{h^*} \phi'\left(h^{-1}(t)\right) h^{-1}(t)\,\mathrm{d}t + o(1) \right], \tag{39}$$

where $o(1) \to 0$ as $\epsilon \to 0$. This completes the proof. $\qquad\square$

## C Proof of Theorem 4.1

In equation 16, we observe that $\hat{\mu}_f$ depends on $\hat{\mu}_c$ and $\hat{\sigma}_c$. As $\hat{\mu}_c$ and $\hat{\sigma}_c$ are not constants but random variables, the procedure used to derive the results for the non-adaptive scheme cannot be directly applied here.

Let us define:

$$T_{n_3} \triangleq \sqrt{n_3} \frac{1}{\sqrt{\mathrm{Var}\left[Y_{n_1+n_2+1}|\hat{\mu}_c\right]}} \left[ \frac{1}{n_3} \sum_{i=n_1+n_2+1}^{n} (Y_i - A_{n_3}) \right], \tag{40}$$

where

$$A_{n_3} \triangleq \mathbb{E}\left[Y_{n_1+n_2+1}|\hat{\mu}_c\right] = \mathbb{P}\left[X_{n_1+n_2+1} \leqslant \hat{\mu}_c\right] = F_X\left(\frac{\hat{\mu}_c - \mu}{\sigma}\right).$$

The conditional variance is

$$\mathrm{Var}\left[Y_{n_1+n_2+1}|\hat{\mu}_c\right] = A_{n_3}(1 - A_{n_3})$$

and

$$\mathbb{E}\left[|Y_{n_1+n_2+1} - A_{n_3}|^3 \mid \hat{\mu}_c\right] \leqslant (1 - A_{n_3})A_{n_3}.$$

From the Berry-Esseen theorem (Vershynin, 2018, Theorem 2.1.3) there exists a universal constant $C$ such that:

$$\sup_{x\in\mathbb{R}} \left| F_{T_{n_3}|\hat{\mu}_c}(x) - \Phi(x) \right| \leqslant \frac{C}{\sqrt{n_3}\sqrt{(1 - A_{n_3})A_{n_3}}}. \tag{41}$$

where $F_{T_{n_3}|\hat{\mu}_c}$ is the conditional cdf of $T_{n_3}$ given $\hat{\mu}_c$, and $\Phi(x)$ is the standard normal cumulative distribution function.

Now we can rewrite equation 40 as:

$$T_{n_3} = \sqrt{n_3} \frac{1}{\sqrt{A_{n_3}(1 - A_{n_3})}} \left[(F_X(\alpha_3) - A_{n_3})\right] \tag{42}$$

Rearranging the terms,

$$\alpha_3 - \frac{\hat{\mu}_c - \mu}{\sigma} = F_X^{-1}\left(\frac{T_{n_3}\left(\sqrt{A_{n_3}(1 - A_{n_3})}\right)}{\sqrt{n_3}}\right) + A_{n_3} - \left(\frac{\hat{\mu}_c - \mu}{\sigma}\right). \tag{43}$$

Let us define

$$H \triangleq \sqrt{n_3}\left(\alpha_3 - \frac{\hat{\mu}_c - \mu}{\sigma}\right) \tag{44}$$

For any $h \in \mathbb{R}$,

$$H \leqslant h \implies \sqrt{n_3}\left(\alpha_3 - \frac{\hat{\mu}_c - \mu}{\sigma}\right) \leqslant h \implies \alpha_3 \leqslant \frac{h}{\sqrt{n_3}} + \frac{\hat{\mu}_c - \mu}{\sigma}.$$

Using equation 42,

$$\{H \leqslant h\} \equiv \left\{T_{n_3} \leqslant \frac{\sqrt{n_3}\left[F_X\left(\frac{h}{\sqrt{n_3}} + \frac{\hat{\mu}_c - \mu}{\sigma}\right) - F_X\left(\frac{\hat{\mu}_c - \mu}{\sigma}\right)\right]}{\sqrt{A_{n_3}(1 - A_{n_3})}}\right\}. \tag{45}$$

Using the Taylor series, we can write,

$$F_X\left(\frac{h}{\sqrt{n_3}} + \frac{\hat{\mu}_c - \mu}{\sigma}\right) - F_X\left(\frac{\hat{\mu}_c - \mu}{\sigma}\right) = f_X\left(\frac{\hat{\mu}_c - \mu}{\sigma}\right)\left(\frac{h}{\sqrt{n_3}}\right) + \mathcal{O}\left(\frac{h^2}{n_3}\right).$$

Hence, equation 45 becomes

$$\{H \leqslant h\} \equiv \left\{T_{n_3} \leqslant \frac{\sqrt{n_3}\left[f_X\left(\frac{\hat{\mu}_c - \mu}{\sigma}\right)\left(\frac{h}{\sqrt{n_3}}\right) + \mathcal{O}\left(\frac{h^2}{n_3}\right)\right]}{\sqrt{A_{n_3}(1 - A_{n_3})}}\right\}. \tag{46}$$

Therefore

$$F_{H|\hat{\mu}_c, \hat{\sigma}_c}(h) = F_{T_{n_3}|\hat{\mu}_c, \hat{\sigma}_c}\left[\frac{\sqrt{n_3}\left[f_X\left(\frac{\hat{\mu}_c - \mu}{\sigma}\right)\left(\frac{h}{\sqrt{n_3}}\right) + \mathcal{O}\left(\frac{h^2}{n_3}\right)\right]}{\sqrt{A_{n_3}(1 - A_{n_3})}}\right].$$

Now from equation 41

$$\sup_{h \in \mathbb{R}}\left|F_{H|\hat{\mu}_c, \hat{\sigma}_c}(h) - \Phi\left(\frac{\sqrt{n_3}\left[f_X\left(\frac{\hat{\mu}_c - \mu}{\sigma}\right)\left(\frac{h}{\sqrt{n_3}}\right) + \mathcal{O}\left(\frac{h^2}{n_3}\right)\right]}{\sqrt{A_{n_3}(1 - A_{n_3})}}\right)\right| \leqslant \frac{C}{\sqrt{n_3}\sqrt{(1 - A_{n_3})A_{n_3}}}. \tag{47}$$

Now, recall the expression for $\hat{\mu}_f$ from equation 16

$$\hat{\mu}_f = \hat{\mu} - F_X^{-1}(F_3) \cdot \hat{\sigma}_c.$$

Since $\alpha_3 = F_X^{-1}(F_3) = \frac{H}{\sqrt{n_3}} + \left(\frac{\hat{\mu}_c - \mu}{\sigma}\right)$, we can write

$$\hat{\mu}_f = \hat{\mu} - \frac{H}{\sqrt{n_3}}\hat{\sigma}_c - \left(\frac{\hat{\mu}_c - \mu}{\sigma}\right)\hat{\sigma}_c$$

and

$$M_f \triangleq \sqrt{n_3}(\mu - \hat{\mu}_f) = H\hat{\sigma}_c + \sqrt{n_3}\left[\frac{(\sigma - \hat{\sigma}_c)(\mu - \hat{\mu}_c)}{\sigma}\right]. \tag{48}$$

Let $\beta \triangleq \sqrt{n_3}\left[\frac{(\sigma - \hat{\sigma}_c)(\mu - \hat{\mu}_c)}{\sigma}\right]$ and $M_f \triangleq \sqrt{n_3}(\mu - \hat{\mu}_f)$, so that

$$M_f = H\hat{\sigma}_c + \beta.$$

The event

$$\{M_f \leqslant m\} \equiv \left\{ H \leqslant \frac{m - \beta}{\hat{\sigma}_c} \right\}. \tag{49}$$

Using this and equation 47, we obtain

$$\sup_{m \in \mathbb{R}} \left| F_{M_f | \hat{\mu}_c, \hat{\sigma}_c}(m) - \Phi \left( \frac{f_X \left( \frac{\hat{\mu}_c - \mu}{\sigma} \right) \left( \frac{m - \beta}{\hat{\sigma}_c} \right)}{\sqrt{A_{n_3}(1 - A_{n_3})}} + \mathcal{O} \left( \frac{(m - \beta)^2}{n_3 \hat{\sigma}_c} \right) \right) \right| \leqslant \frac{C}{\sqrt{n_3} \sqrt{(1 - A_{n_3}) A_{n_3}}}. \tag{50}$$

Now, we define $\zeta \triangleq \frac{f_X \left( \frac{\hat{\mu}_c - \mu}{\sigma} \right) \beta}{\hat{\sigma}_c \sqrt{A_{n_3}(1 - A_{n_3})}} - \mathcal{O} \left( \frac{(m - \beta)^2}{n_3 \hat{\sigma}_c^2} \right)$. For any fixed $m \in \mathbb{R}$, equation 50 can be written as:

$$\left| F_{M_f | \hat{\mu}_c, \hat{\sigma}_c}(m) - \Phi \left( \frac{f_X \left( \frac{\hat{\mu}_c - \mu}{\sigma} \right) m}{\hat{\sigma}_c \sqrt{A_{n_3}(1 - A_{n_3})}} - \zeta \right) \right| \leqslant \frac{C}{\sqrt{n_3} \sqrt{(1 - A_{n_3}) A_{n_3}}}. \tag{51}$$

Using the triangle inequality

$$\left| F_{M_f | \hat{\mu}_c, \hat{\sigma}_c}(m) - \Phi \left( \frac{f_X(0) m}{\sigma/2} \right) \right| \leqslant \left| F_{M_f | \hat{\mu}_c, \hat{\sigma}_c}(m) - \Phi \left( \frac{f_X \left( \frac{\hat{\mu}_c - \mu}{\sigma} \right) m}{\hat{\sigma}_c \sqrt{A_{n_3}(1 - A_{n_3})}} - \zeta \right) \right|$$
$$+ \left| \Phi \left( \frac{f_X \left( \frac{\hat{\mu}_c - \mu}{\sigma} \right) m}{\hat{\sigma}_c \sqrt{A_{n_3}(1 - A_{n_3})}} - \zeta \right) - \Phi \left( \frac{f_X(0) m}{\sigma/2} \right) \right|. \tag{52}$$

Since $\Phi(x)$ is Lipschitz continuous, we can write

$$\left| \Phi \left( \frac{f_X \left( \frac{\hat{\mu}_c - \mu}{\sigma} \right) m}{\hat{\sigma}_c \sqrt{A_{n_3}(1 - A_{n_3})}} - \zeta \right) - \Phi \left( \frac{f_X(0) m}{\sigma/2} \right) \right| \leqslant \frac{1}{\sqrt{2\pi}} \left| \left( \frac{f_X \left( \frac{\hat{\mu}_c - \mu}{\sigma} \right) m}{\hat{\sigma}_c \sqrt{A_{n_3}(1 - A_{n_3})}} - \zeta \right) - \frac{f_X(0) m}{\sigma/2} \right|. \tag{53}$$

Using equation 53 and equation 51 in equation 52, we can write

$$\left| F_{M_f | \hat{\mu}_c, \hat{\sigma}_c}(m) - \Phi \left( \frac{f_X(0) m}{\sigma/2} \right) \right| \leqslant \frac{C}{\sqrt{n_3} \sqrt{(1 - A_{n_3}) A_{n_3}}} + \frac{1}{\sqrt{2\pi}} \left| \left( \frac{f_X \left( \frac{\hat{\mu}_c - \mu}{\sigma} \right) m}{\hat{\sigma}_c \sqrt{A_{n_3}(1 - A_{n_3})}} - \zeta \right) - \frac{f_X(0) m}{\sigma/2} \right| \tag{54}$$

where

$$\zeta = \frac{f_X \left( \frac{\hat{\mu}_c - \mu}{\sigma} \right) \sqrt{n_3} \left( (\sigma - \hat{\sigma}_c) (\mu - \hat{\mu}_c) \right)}{\hat{\sigma}_c \cdot \sigma \sqrt{A_{n_3}(1 - A_{n_3})}} - \mathcal{O} \left( \frac{(m - \beta)^2}{n_3 \hat{\sigma}_c^2} \right).$$

We apply the Jensen's inequality:

$$\mathop{\mathbb{E}}_{\hat{\mu}_c, \hat{\sigma}_c} \left( \left| F_{M_f | \hat{\mu}_c, \hat{\sigma}_c}(m) - \Phi \left( \frac{f_X(0) m}{\sigma/2} \right) \right| \right) \geqslant \left| \mathop{\mathbb{E}}_{\hat{\mu}_c, \hat{\sigma}_c} \left( F_{M_f | \hat{\mu}_c, \hat{\sigma}_c}(m) - \Phi \left( \frac{f_X(0) m}{\sigma/2} \right) \right) \right|.$$

Therefore

$$\mathop{\mathbb{E}}_{\hat{\mu}_c, \hat{\sigma}_c} \left| F_{M_f | \hat{\mu}_c, \hat{\sigma}_c}(m) - \Phi \left( \frac{f_X(0) m}{\sigma/2} \right) \right| \geqslant \left| F_{M_f}(m) - \Phi \left( \frac{f_X(0) m}{\sigma/2} \right) \right|.$$

Now, equation 54 becomes

$$\left| F_{M_f}(m) - \Phi\left(\frac{f_X(0)m}{\sigma/2}\right) \right| \leqslant \frac{C}{\sqrt{n_3}\sqrt{(1-A_{n_3})A_{n_3}}} + \underset{\hat{\mu}_c, \hat{\sigma}_c}{\mathbb{E}} \left( \frac{1}{\sqrt{2\pi}} \cdot \left| \left( \frac{f_X\left(\frac{\hat{\mu}-\mu}{\sigma}\right) m}{\hat{\sigma}_c\sqrt{A_{n_3}(1-A_{n_3})}} - \zeta \right) - \frac{f_X(0)m}{\sigma/2} \right| \right).$$

$$(55)$$

Now our goal is to show that the R.H.S. of equation 55 converges to zero as $n \to \infty$. To achieve this, we first establish the convergence of $\sqrt{n_3}\left((\mu - \hat{\mu}_c)(\sigma - \hat{\sigma}_c)\right)$. The following lemma provides this convergence result.

**Lemma C.1.** *Let $\hat{\mu}_c$ and $\hat{\sigma}_c$ be estimates of $\mu$ and $\sigma$ using the non-adaptive protocol described in Sec. 3. Then*

$$P\left[ \sqrt{n_3}\,|\sigma - \hat{\sigma}_c|\,|\mu - \hat{\mu}_c| \geqslant \frac{8|\mu\sigma|(\log n_3)^2/\sqrt{n_3}}{\left(1 - 2(\log n_3)/\sqrt{n_3}\right)^2} \right] \leqslant 4\exp\left(\frac{-\tilde{C}_1\log n_3}{3}\right)\left[1 + \exp\left(\frac{\eta\log n_3}{3}\right)\right].$$

*Where $\tilde{C}_1$ is a constant that depends on $F_X$, $\theta_i$, $\sigma$, and $\mu$, but independent of $n$. As $n \to \infty$,*

$$\sqrt{n_3}\,(\mu - \hat{\mu}_c)(\sigma - \hat{\sigma}_c) \xrightarrow{p} 0.$$

Please note that as $n \to \infty$, we have $\sqrt{A_{n_3}(1 - A_{n_3})} \xrightarrow{\text{a.s.}} 1/2$, $\hat{\mu}_c \xrightarrow{\text{a.s.}} \mu$, $\hat{\sigma}_c \xrightarrow{\text{a.s.}} \sigma$, and from Lemma C.1, $\sqrt{n_3}\left((\mu - \hat{\mu}_c)(\sigma - \hat{\sigma}_c)\right) \xrightarrow{\text{p}} 0$. Consequently,

$$\zeta \xrightarrow{p} 0,$$

and

$$\frac{f_X\left(\frac{\hat{\mu}-\mu}{\sigma}\right)}{\left(\hat{\sigma}_c\sqrt{A_{n_3}(1-A_{n_3})}\right)} \xrightarrow{\text{a.s.}} \frac{f_X(0)m}{\sigma/2}.$$

Thus, the R.H.S. of equation 55 converges to zero as $n \to \infty$, and we can write:

$$M_f = \sqrt{n}\,(\mu_f - \mu) \xrightarrow{d} \mathcal{N}\left(0, \frac{\sigma^2}{4f_X(0)^2}\right).$$

Finally, this gives us

$$\lim_{n\to\infty} n \cdot \text{MSE}(\hat{\mu}_f) = \frac{\sigma^2}{4f_X(0)^2}.$$

This concludes the proof $\qquad\qquad\qquad\qquad\qquad\qquad\qquad\qquad\qquad\qquad\qquad\qquad\qquad\qquad\qquad\square$

## D  Proof of Theorem 4.3

To establish a lower bound on the mean-squared error (MSE) of our estimator, we employ the Van Trees inequality (Gill & Levit, 1995; Van Trees, 2004), which states that for any estimator,

$$\mathbb{E}\left[(\hat{\mu} - \mu)^2\right] \geqslant \frac{1}{\mathbb{E}_g\left[\mathbf{I}_\mu\right] + I_0} \tag{56}$$

where $g(\cdot)$ is any prior density over the parameter $\mu$ that satisfies the regularity conditions. The expectation on the left hand side is with respect to the joint density $g(\mu)f_{X,\mu,\sigma}(x)$ on $(\mu, x)$, while $\mathbb{E}_g[\cdot]$ denotes expectation with respect to $g(\mu)$. Also, $I_0$ and $\mathbf{I}_\mu$ are defined as:

$$I_0 = \mathbb{E}_g\left[\left(\frac{g'(\mu)}{g(\mu)}\right)^2\right] \quad \text{and} \quad \mathbf{I}_\mu = \mathbb{E}\left[\left(\frac{\partial}{\partial\mu}\log P(\underline{Y}\mid\mu)\right)^2\right]$$

respectively. Here $g'(\mu)$ is the the derivative of $g(\mu)$ with respect to $\mu$ and $\underline{Y} = [Y_1, Y_2, \ldots, Y_n]$, where each $Y_i$ is the 1-bit output of user $i$. Moreover, $P(\underline{Y}\mid\mu)$ denotes the joint distribution of $Y_1, \ldots, Y_n$ given $\mu$. To keep notation simple, we have not explicitly mentioned the dependence on $\sigma$.

Using the chain rule of Fisher information (Zamir, 1998),

$$\mathbf{I}_\mu = \sum_{i=1}^{n} I_{\mu,i},$$ (57)

where the Fisher information

$$I_{\mu,i} = \mathbb{E}\left[\left(\frac{\partial}{\partial \mu} \log P\big(Y_i \mid Y_1, \ldots, Y_{i-1}, \mu\big)\right)^2\right].$$

By definition of an adaptive protocol, $Y_i$ can depend only on $X_i$ and $Y_1, \ldots, Y_{i-1}$. Moreover, $X_i$ is independent of $Y_1, \ldots, Y_{i-1}$. Therefore, we can write

$$Y_i = \begin{cases} 1, & \text{if } X_i \in \mathcal{A}_i \\ 0, & \text{otherwise.} \end{cases}$$

where each $\mathcal{A}_i$ is a Borel measurable set that can depend only on $Y_1, \ldots, Y_{i-1}$ and therefore independent of $X_i$.

**Lemma D.1.** *Let $\mathcal{S}$ be a Borel measurable set independent of $X_i$, and define $Y_i$ as:*

$$Y_i = \begin{cases} 1 & \text{if } X_i \in \mathcal{S}, \\ 0 & \text{otherwise.} \end{cases}$$

*Then, for any $\delta > 0$, the Fisher information $I_{\mu,i}$ of $Y_i$ with respect to $\mu$ is upper bounded by:*

$$I_{\mu,i} \leqslant \left(\frac{2 f_X(0)}{\sigma}\right)^{2+\delta}.$$

Using the fact that equation 58 holds for every $\delta > 0$, and equation 57,

$$\mathbf{I}_\mu \leqslant n \left(\frac{2 f_X(0)}{\sigma}\right)^2.$$

Substituting this upper bound into the Van Trees inequality equation 56 provides the final lower bound on the MSE:

$$\mathbb{E}\left[(\hat{\mu} - \mu)^2\right] \geqslant \frac{1}{n \left(\frac{2 f_X(0)}{\sigma}\right)^2 + I_0} = \frac{1}{n \left[\left(\frac{2 f_X(0)}{\sigma}\right)^2 + \frac{I_0}{n}\right]}.$$

As $n \to \infty$, the term $\frac{I_0}{n}$ approaches zero (since $I_0$ is assumed to be bounded). Consequently, we obtain:

$$\lim_{n \to \infty} n \cdot \mathbb{E}\left[(\hat{\mu} - \mu)^2\right] \geqslant \frac{\sigma^2}{4 f_X(0)^2}.$$ (58)

This concludes the proof. □

## E    Proof of Theorem 3.3

We analyze bias and variance separately.

### E.0.1 Bias

Clearly,

$$\mathbb{E}[\hat{\mu}_{\mathrm{MT}}] = \hat{\mu}_{\Theta}.$$

We now bound the term $|\mathbb{E}[X] - \mathbb{E}[\hat{\mu}_{\mathrm{MT}}]|$.

Recall that

$$\theta_j = \mu + j\Delta\sigma, \qquad \bar{\theta}_j = \frac{\theta_j + \theta_{j+1}}{2}, \qquad \Delta = \frac{2h}{m}.$$

Using the representation of $\hat{\mu}_{\mathrm{MT}}$ and separating the negative and positive intervals, we write

$$
\begin{aligned}
|\mathbb{E}[X] - \mathbb{E}[\hat{\mu}_{\mathrm{MT}}]| &= \left| -\sum_{j=-m}^{-1} \left( \int_{\theta_j}^{\theta_{j+1}} F_X(z)\, dz - F_X(\bar{\theta}_j)\Delta \right) \right. \\
&\qquad \left. + \sum_{j=1}^{m} \left( \int_{\theta_{j-1}}^{\theta_j} (1 - F_X(z))\, dz - (1 - F_X(\bar{\theta}_j))\Delta \right) \right| \\
&= \left| -\sum_{j=-m}^{-1} \int_{\theta_j}^{\theta_{j+1}} \left( F_X(z) - F_X(\bar{\theta}_j) \right) dz \right. \\
&\qquad \left. + \sum_{j=1}^{m} \int_{\theta_{j-1}}^{\theta_j} \left( (1 - F_X(z)) - (1 - F_X(\bar{\theta}_j)) \right) dz \right| \qquad (59) \\
&\leqslant \sum_{j=-m}^{-1} \int_{\theta_j}^{\theta_{j+1}} |F_X(z) - F_X(\bar{\theta}_j)|\, dz + \sum_{j=1}^{m} \int_{\theta_{j-1}}^{\theta_j} |F_X(\bar{\theta}_j) - F_X(z)|\, dz. \qquad (60)
\end{aligned}
$$

Since $f_X(t) \leqslant L$ for all $t$, the CDF $F_X$ is $L$-Lipschitz in its argument:

$$|F_X(u) - F_X(v)| \leqslant L|u - v|.$$

In our estimator, the argument of $F_X$ is the normalized variable $(z - \mu)/\sigma$. Thus,

$$\left| F_X\left( \frac{z - \mu}{\sigma} \right) - F_X\left( \frac{\bar{\theta}_j - \mu}{\sigma} \right) \right| \leqslant L \left| \frac{z - \bar{\theta}_j}{\sigma} \right|.$$

For $z \in [\theta_j, \theta_{j+1}]$, we have

$$|z - \bar{\theta}_j| \leqslant \frac{\theta_{j+1} - \theta_j}{2} = \frac{\Delta\sigma}{2}.$$

Substituting into the Lipschitz bound gives

$$\left| F_X\left( \frac{z - \mu}{\sigma} \right) - F_X\left( \frac{\bar{\theta}_j - \mu}{\sigma} \right) \right| \leqslant \frac{L\Delta}{2}.$$

Therefore,

$$\int_{\theta_j}^{\theta_{j+1}} |F_X(z) - F_X(\bar{\theta}_j)|\, dz \leqslant \frac{L\Delta^2}{2},$$

and the same bound holds for the positive intervals. Summing over all $2m$ bins yields

$$|\mathbb{E}[X] - \mathbb{E}[\hat{\mu}_{\mathrm{MT}}]| \leqslant Lm\Delta^2.$$

### E.0.2 Variance

For every $j \in \{-m, \ldots, -1, 1, \ldots, m\}$,

$$\mathrm{Var}(\bar{I}_j) = \frac{F_X(\bar{\theta}_j)\big(1 - F_X(\bar{\theta}_j)\big)}{K}.$$

Therefore,

$$\mathrm{Var}(\hat{\mu}_{MT}) = \sum_{\substack{j=-m \\ j \neq 0}}^{m} \Delta^2 \, \mathrm{Var}(\bar{I}_j) = \frac{\Delta^2}{K} \sum_{\substack{j=-m \\ j \neq 0}}^{m} F_X(\bar{\theta}_j)\big(1 - F_X(\bar{\theta}_j)\big). \tag{61}$$

Since $\theta_j = \mu + j\Delta\sigma$, the term

$$F_X(\bar{\theta}_j)\big(1 - F_X(\bar{\theta}_j)\big)$$

is evaluated at points forming a uniform grid of spacing $\Delta\sigma$. Hence

$$\sum_{\substack{j=-m \\ j \neq 0}}^{m} F_X(\bar{\theta}_j)\big(1 - F_X(\bar{\theta}_j)\big)\Delta$$

is a Riemann sum for the integral

$$\int_{-\infty}^{\infty} F_X(x)\big(1 - F_X(x)\big)\,dx,$$

If we choose the parameters such that $\Delta \to 0$ and $\theta_m = -\theta_{-m} \to \infty$, then

$$\mathrm{Var}(\hat{\mu}_{MT}) = \frac{\Delta}{K} \left( \int_{-\infty}^{\infty} F_X(x)\big(1 - F_X(x)\big)\,dx + o(1) \right).$$

This completes the proof. $\qquad\square$

## F  Bounding the Total Variation Distance using the Squared Hellinger Distance

**Lemma F.1.** *For any pair of discrete distributions[7] $P_Y^+, P_Y^-$ defined on a common alphabet,*

$$\mathrm{TV}(P_Y^-, P_Y^+) \;\leqslant\; \sqrt{1 - \Big(1 - H^2(P_Y^-, P_Y^+)\Big)^2}.$$

*Proof.* The following relationship between the squared Hellinger distance and the Bhattacharyya coefficient is standard, but we nevertheless provide a proof for completeness. For discrete distributions $P_Y^+, P_Y^-$, we have

$$H^2\left(P_Y^-, P_Y^+\right) = \tfrac{1}{2} \sum_y \left( \sqrt{p_Y^-(y)} - \sqrt{p_Y^+(y)} \right)^2$$

$$= 1 - \rho(P_Y^-, P_Y^+), \tag{62}$$

where

$$\rho(P_Y^-, P_Y^+) \triangleq \sum_y \sqrt{p_Y^-(y)\, p_Y^+(y)}$$

is the Bhattacharyya coefficient.

Let $u(y) := \sqrt{p_Y^-(y)}$ and $v(y) := \sqrt{p_Y^+(y)}$. Then

$$\left| p_Y^-(y) - p_Y^+(y) \right| = |(u(y) - v(y))(u(y) + v(y))|,$$

---

[7]An identical approach works for continuous distributions as well.

and by the Cauchy–Schwarz inequality,

$$\sum_y \left| p_Y^-(y) - p_Y^+(y) \right| \;\leqslant\; \left( \sum_y (u(y) - v(y))^2 \right)^{1/2} \left( \sum_y (u(y) + v(y))^2 \right)^{1/2}.$$

Simplifying,

$$\sum_y \left| p_Y^-(y) - p_Y^+(y) \right| \leqslant \left( 2\left(1 - \rho(P_Y^-, P_Y^+)\right) \right)^{1/2} \left( 2\left(1 + \rho(P_Y^-, P_Y^+)\right) \right)^{1/2}$$

$$= 2\sqrt{1 - \rho(P_Y^-, P_Y^+)^2}. \tag{63}$$

Therefore,

$$\mathrm{TV}(P_Y^-, P_Y^+) \;\leqslant\; \sqrt{1 - \rho(P_Y^-, P_Y^+)^2}.$$

Finally, using equation 62 yields

$$\mathrm{TV}(P_Y^-, P_Y^+) \;\leqslant\; \sqrt{1 - \left(1 - H^2(P_Y^-, P_Y^+)\right)^2},$$

which gives us the lemma. $\qquad\square$

## G  Proofs of Lemmas Appearing in the Proof of Theorem B.1

### G.1  Proof of lemma B.2

We now define a generalized squared Hellinger distance. Let $\mathcal{Z}$ be a finite set, $\pi_i : \mathbb{R} \to \mathcal{Z}$ a (possibly randomized) function, and $f_1, f_2$ be non-negative functions. Then

$$H^2(\pi_i(X); f_1(x), f_2(x)) \triangleq \tfrac{1}{2} \sum_{z \in \mathcal{Z}} \left( \sqrt{\int_{-\infty}^{\infty} f_1(x) \Pr[\pi_i(X) = z \mid X = x] \mathrm{d}x} - \sqrt{\int_{-\infty}^{\infty} f_2(x) \Pr[\pi_i(X) = z \mid X = x] \mathrm{d}x} \right)^2,$$

which reduces to the usual squared Hellinger distance when $f_1, f_2$ are valid densities.

Similarly, if $\pi_i$ is a function with output alphabet $\mathcal{Z} = \{0, 1\}$, the generalized total variation distance is

$$\mathrm{TV}(\pi_i(X); f_1(x), f_2(x)) \triangleq \tfrac{1}{2} \sum_{z=0}^{1} \left| \int_{-\infty}^{\infty} f_1(x) \Pr[\pi_i(X) = z \mid X = x] \mathrm{d}x - \int_{-\infty}^{\infty} f_2(x) \Pr[\pi_i(X) = z \mid X = x] \mathrm{d}x \right|.$$

Now we recall our definitions:

$$f_X(x + \epsilon) = \int_0^{a_\epsilon^*} g_+(x, a) \, \mathrm{d}a \quad \text{and} \quad f_X(x - \epsilon) = \int_0^{a_\epsilon^*} g_-(x, a) \, \mathrm{d}a.$$

Therefore,

$$\frac{1}{\sigma} f_X\left( \frac{x - \mu_-}{\sigma} \right) = \frac{1}{\sigma} f_X\left( \frac{x - (\mu - \epsilon\sigma)}{\sigma} \right) = \frac{1}{\sigma} f_X\left( \frac{x - \mu}{\sigma} + \epsilon \right) = \int_0^{a_\epsilon^*} \frac{1}{\sigma} g_+\left( \frac{x - \mu}{\sigma}, a \right) \, \mathrm{d}a \tag{64}$$

and likewise,

$$\frac{1}{\sigma} f_X\left( \frac{x - \mu_+}{\sigma} \right) = \frac{1}{\sigma} f_X\left( \frac{x - (\mu + \epsilon\sigma)}{\sigma} \right) = \int_0^{a_\epsilon^*} \frac{1}{\sigma} g_-\left( \frac{x - \mu}{\sigma}, a \right) \, \mathrm{d}a. \tag{65}$$

Using equation 64, equation 65, and (Cai & Wei, 2022a, Lemma 2),

$$H^2(P^+_{Y_i}; P^-_{Y_i}) = H^2\left(\pi_i(x); \frac{1}{\sigma}f_X\left(\frac{x-\mu_+}{\sigma}\right), \frac{1}{\sigma}f_X\left(\frac{x-\mu_-}{\sigma}\right)\right)$$

$$\leqslant \int_0^{a^*_\epsilon} H^2\left(\pi_i(x); \frac{1}{\sigma}g_+\left(\frac{x-\epsilon\sigma}{\sigma}, a\right), \frac{1}{\sigma}g_-\left(\frac{x-\epsilon\sigma}{\sigma}, a\right)\right) \mathrm{d}a$$

$$\leqslant \int_0^{a^*_\epsilon} \left(\frac{\sqrt{R_a}-1}{\sqrt{R_a}+1}\right) \mathrm{TV}\left(\pi_i(x); \frac{1}{\sigma}g_+\left(\frac{x-\mu}{\sigma}, a\right), \frac{1}{\sigma}g_-\left(\frac{x-\mu}{\sigma}, a\right)\right) \mathrm{d}a. \tag{66}$$

Now for any $R_a \geqslant 1$ that satisfies

$$\frac{1}{R_a} \leqslant \frac{g_+(x)}{g_-(x)} \leqslant R_a \quad \text{for all } x \text{ such that } g_-(x) > 0.$$

We can write,

$$\mathrm{TV}\left(\pi_i(x); \frac{1}{\sigma}g_+\left(\frac{x-\mu}{\sigma}, a\right), \frac{1}{\sigma}g_-\left(\frac{x-\mu}{\sigma}, a\right)\right)$$

$$\leqslant \frac{1}{2}\int_{-\infty}^{\infty} \sum_{z=0}^{1} \Pr[\pi_i(x) = z | X = x]\left|\frac{1}{\sigma}g_+\left(\frac{x-\mu}{\sigma}, a\right) - \frac{1}{\sigma}g_-\left(\frac{x-\mu}{\sigma}, a\right)\right| \mathbf{1}_{\{\frac{x-\mu}{\sigma} \in A_\epsilon(a)\}} \mathrm{d}x$$

$$= \frac{1}{2}\int_{-\infty}^{\infty} \left|\frac{1}{\sigma}g_+\left(\frac{x-\mu}{\sigma}, a\right) - \frac{1}{\sigma}g_-\left(\frac{x-\mu}{\sigma}, a\right)\right| \mathbf{1}_{\{\frac{x-\mu}{\sigma} \in A_\epsilon(a)\}} \mathrm{d}x$$

$$= \frac{1}{2}\int_{t \in A_\epsilon(a)} |g_+(t, a) - g_-(t, a)| \mathrm{d}t$$

$$= \frac{1}{2}|A_\epsilon(a)| \leqslant \frac{1}{2}(2x_a) = x_a.$$

Where the third step follows from a change of variable, and the penultimate step uses the definition of $x_a$. Now using this bound on TV in equation 66, we have,

$$H^2(P^+_{Y_i}; P^-_{Y_i}) \leqslant \int_0^{a^*_\epsilon} \left(\frac{\sqrt{R_a}-1}{\sqrt{R_a}+1}\right) x_a, \mathrm{d}a. \tag{67}$$

This completes the proof. $\qquad\square$

### G.2 Proof of Lemma B.3

Since $f_X(x) = e^{-\phi(x)}$, we have

$$\frac{f_X(x+\epsilon)}{f_X(x-\epsilon)} = e^{-\phi(x+\epsilon)+\phi(x-\epsilon)} = e^{-\delta(x,\epsilon)},$$

where

$$\delta(x,\epsilon) \triangleq \phi(x+\epsilon) - \phi(x-\epsilon).$$

Therefore,

$$R_a = \sup_{x \in A_\epsilon(a)} e^{-\delta(x,\epsilon)} = e^{-\inf_{x \in A_\epsilon(a)} \delta(x,\epsilon)}.$$

From the strict convexity of $\phi$ and the definition of $\delta$:

- $\delta(x,\epsilon)$ is strictly increasing on $[0,\infty)$,

- Evenness of $\phi$ implies $\delta(x,\epsilon)$ is odd: $\delta(-x,\epsilon) = -\delta(x,\epsilon)$.

Thus $\delta(x, \epsilon)$ changes sign exactly once at $x = 0$, being negative for $x < 0$ and positive for $x > 0$. Since $A_\epsilon(a)$ is symmetric, the smallest value of $\delta(x, \epsilon)$ over $A_\epsilon(a)$ occurs at the left endpoint, which equals $-|\delta(x_a(\epsilon), \epsilon)|$, where $x_a(\epsilon) = \sup A_\epsilon(a)$.

Consequently,

$$\inf_{x \in A_\epsilon(a)} \delta(x, \epsilon) = -|\delta(x_a(\epsilon), \epsilon)|,$$

and

$$R_a = e^{-\inf_{x \in A_\epsilon(a)} \delta(x, \epsilon)} = e^{|\delta(x_a(\epsilon), \epsilon)|}.$$

$R_a = e^{\delta(x_a(\epsilon), \epsilon)}$, so

$$\frac{\sqrt{R_a} - 1}{\sqrt{R_a} + 1} = \tanh\left(\frac{\delta(x_a(\epsilon), \epsilon)}{4}\right).$$

Since $\tanh(t) \leqslant t$ for all $t \geqslant 0$,

$$\frac{\sqrt{R_a} - 1}{\sqrt{R_a} + 1} x_a(\epsilon) \leqslant \frac{\delta(x_a(\epsilon), \epsilon)}{4} x_a(\epsilon). \tag{68}$$

Since $\phi$ is convex and differentiable, we can use the first-order condition for convexity to get

$$2\epsilon \cdot \phi'(x - \epsilon) \leqslant \phi(x + \epsilon) - \phi(x - \epsilon) \leqslant 2\epsilon \cdot \phi'(x + \epsilon).$$

Therefore,

$$\delta(x_a(\epsilon), \epsilon) \leqslant 2\epsilon\, \phi'(x_a(\epsilon) + \epsilon),$$

which yields

$$\frac{\delta(x_a(\epsilon), \epsilon)}{4} x_a(\epsilon) \leqslant \frac{\epsilon\, \phi'(x_a(\epsilon) + \epsilon)\, x_a(\epsilon)}{2}.$$

Therefore, equation 68 can be written as,

$$\frac{\sqrt{R_a} - 1}{\sqrt{R_a} + 1} x_a(\epsilon) \leqslant \frac{\epsilon}{2} \phi'(x_a(\epsilon) + \epsilon)\, x_a(\epsilon).$$

Substituting this pointwise bound into the integral directly yields the result. $\qquad\square$

## H   Proofs of Lemmas C.1 and D.1

### H.1   Proof of Lemma C.1

To prove the convergence of $\sqrt{n_3}(\mu - \hat{\mu}_c)(\sigma - \hat{\sigma}_c)$, we first state two key Lemmas that show the convergence of $(\hat{\mu}_c - \mu)$ and $(\hat{\sigma}_c - \sigma)$, respectively.

**Lemma H.1.** *Let $\hat{\mu}_c$ be the estimate of $\mu$ using the non-adaptive scheme described in Sec. 3. Then,*

$$\Pr\left(|\hat{\mu}_c - \mu| \geqslant |\mu|\left(\frac{4 \log n_3}{\sqrt{n_3} - 2 \log n_3}\right)\right) \leqslant 2 \exp\left(\frac{-\tilde{C}_1 \log n_3}{3}\right)\left[1 + \exp\left(\frac{-\eta \log n_3}{3}\right)\right], \tag{69}$$

*Here, $\tilde{C}_1$ and $\eta$ depend only on the underlying distribution class and are independent of $n_1, n_2, n_3$.*

**Lemma H.2.** *Let $\hat{\sigma}_c$ be the estimate of $\sigma$ using the non-adaptive scheme described in Sec. 3. Then,*

$$\Pr\left[|\sigma - \hat{\sigma}_c| \geqslant |\sigma|\left(\frac{2 \log n_3}{\sqrt{n_3} - 2 \log n_3}\right)\right] \leqslant 2 \exp\left(\frac{-\tilde{C}_1 \log n_3}{3}\right)\left[1 + \exp\left(\frac{-\eta \log n_3}{3}\right)\right]. \tag{70}$$

*As before, the constants $\tilde{C}_1$ and $\eta$ depend only on the distribution and not on $n_1, n_2, n_3$.*

Since the right-hand sides in Lemmas H.1 and H.2 converge to zero as $n_3 \to \infty$, both $\hat{\mu}_c$ and $\hat{\sigma}_c$ are consistent *in probability* under the non-adaptive scheme. The proof of these two lemmas is given in Appendix I.

Now we begin with some definitions. Let us define the events:

$$E_1 \triangleq \left\{ |\sigma - \hat{\sigma}_c| \geqslant |\sigma| \left( \frac{2 \log n_3}{\sqrt{n_3} - 2 \log n_3} \right) \right\},$$

$$E_2 \triangleq \left\{ |\mu - \hat{\mu}_c| \geqslant |\mu| \left( \frac{4 \log n_3}{\sqrt{n_3} - 2 \log n_3} \right) \right\}.$$

If $A$, $B$, $C$, and $D$ are random variables with $A < B$ and $C < D$, then $AC < BD$. Thus,

$$\{AC \geqslant BD\} \subset \{A \geqslant B\} \cup \{C \geqslant D\},$$

and by the union bound,

$$P(AC \geqslant BD) \leqslant P(A \geqslant B) + P(C \geqslant D).$$

Using this, we obtain

$$\Pr \left[ \sqrt{n_3} \, |\sigma - \hat{\sigma}_c| \, |\mu - \hat{\mu}_c| \geqslant \frac{8|\mu\sigma|(\log n_3)^2/\sqrt{n_3}}{\left(1 - 2(\log n_3)/\sqrt{n_3}\right)^2} \right] \leqslant P[E_1] + P[E_2].$$

Using the bounds for $P(E_1)$ and $P(E_2)$ from Lemmas H.1 and H.2, we obtain:

$$\Pr \left[ \sqrt{n_3} \, |\sigma - \hat{\sigma}_c| \, |\mu - \hat{\mu}_c| \geqslant \frac{8|\mu\sigma|(\log n_3)^2/\sqrt{n_3}}{\left(1 - 2(\log n_3)/\sqrt{n_3}\right)^2} \right] \leqslant 4 \exp \left( \frac{-\tilde{C}_1 \log n_3}{3} \right) \left[ 1 + \exp \left( \frac{-\eta \log n_3}{3} \right) \right].$$

As $n_3 \to \infty$, this probability approaches zero, implying convergence *in probability* for the non-adaptive scheme. □

## H.2 Proof of Lemma D.1

Let $\mathcal{S}$ be a Borel measurable set, and define $Y_i$ as:

$$Y_i = \begin{cases} 1 & \text{if } X_i \in \mathcal{S}, \\ 0 & \text{otherwise.} \end{cases}$$

Now the Fisher information for a parameter $\mu$ is defined as:

$$I_\mu = \mathbb{E} \left[ \left( \frac{\partial}{\partial \mu} \log P(Y_i | \mu) \right)^2 \right].$$

The probability that $Y_i = 1$ is:

$$P(Y_i = 1 | \mu) = \int_{\mathcal{S}} f_{X,\mu,\sigma}(x) dx.$$

By regularity of the Lebesgue measure, we can approximate any Borel measurable set $\mathcal{S}$ using a finite number of disjoint intervals:

$$\mathcal{C} = \bigcup_{j=1}^{k} [a_j^-, a_j^+],$$

with

$$-\infty \leqslant a_1^- < a_1^+ < a_2^- < a_2^+ < \cdots < a_k^+ \leqslant \infty,$$

such that

$$\int_{\mathcal{S} \Delta \mathcal{C}} dx \leqslant \epsilon',$$

where $\Delta$ denotes symmetric difference.

Therefore,

$$P(Y_i = 1|\mu) = \sum_{j=1}^{k} \int_{a_j^-}^{a_j^+} \frac{1}{\sigma} f_X\left(\frac{x - \mu}{\sigma}\right) dx + g_k(\epsilon'),$$

where $g_k(\epsilon')$ is a quantity that can be made arbitrarily close to $0$ by choosing $k$ large enough.

Using the substitution $u = \frac{x-\mu}{\sigma}$, $x = \sigma u + \mu$, and $dx = \sigma du$, the integral becomes

$$P(Y_i = 1|\mu) = \sum_{j=1}^{k} \int_{\frac{a_j^- - \mu}{\sigma}}^{\frac{a_j^+ - \mu}{\sigma}} f_X(u) du + g_k(\epsilon').$$

Recall that $F_X$ as the CDF corresponding to $f_X$. Then

$$P(Y_i = 1|\mu) = \sum_{j=1}^{k} \left[ F_X\left(\frac{a_j^+ - \mu}{\sigma}\right) - F_X\left(\frac{a_j^- - \mu}{\sigma}\right) \right] + g_k(\epsilon').$$

Similarly,

$$P(Y_i = 0|\mu) = 1 - P(Y_i = 1|\mu).$$

Taking the derivative of $P(Y_i = 1|\mu)$ w.r.t. $\mu$, we obtain

$$\frac{\partial}{\partial \mu} P(Y_i = 1|\mu) = \sum_{j=1}^{k} \frac{\partial}{\partial \mu} \left[ F_X\left(\frac{a_j^+ - \mu}{\sigma}\right) - F_X\left(\frac{a_j^- - \mu}{\sigma}\right) \right] + g_k'(\epsilon').$$

Using the chain rule,

$$\frac{\partial}{\partial \mu} F_X\left(\frac{a_j^\pm - \mu}{\sigma}\right) = -\frac{1}{\sigma} f_X\left(\frac{a_j^\pm - \mu}{\sigma}\right).$$

Thus:

$$\frac{\partial}{\partial \mu} P(Y_i = 1|\mu) = -\frac{1}{\sigma} \sum_{j=1}^{k} \left[ f_X\left(\frac{a_j^+ - \mu}{\sigma}\right) - f_X\left(\frac{a_j^- - \mu}{\sigma}\right) \right] + g_k'(\epsilon').$$

Since $Y_i$ is a binary random variable, the Fisher information expands as:

$$I_{\mu,i} = P(Y_i = 1|\mu) \left(\frac{\partial}{\partial \mu} \log P(Y_i = 1|\mu)\right)^2 + P(Y_i = 0|\mu) \left(\frac{\partial}{\partial \mu} \log P(Y_i = 0|\mu)\right)^2 + h_k(\epsilon').$$

for some $h_k(\epsilon')$ that can be made arbitrarily close to $0$ by choosing $k$ large enough.

By substituting the probabilities and derivatives, and then assuming:

$$\Delta F_j(\mu) = F_X\left(\frac{a_j^+ - \mu}{\sigma}\right) - F_X\left(\frac{a_j^- - \mu}{\sigma}\right),$$

$$\Delta f_j(\mu) = \frac{1}{\sigma} \left[ f_X\left(\frac{a_j^+ - \mu}{\sigma}\right) - f_X\left(\frac{a_j^- - \mu}{\sigma}\right) \right],$$

we write

$$P(Y_i = 1|\mu) = \sum_{j=1}^{k} \Delta F_j(\mu) + g_k(\epsilon'), \quad P(Y_i = 0|\mu) = 1 - \sum_{j=1}^{k} \Delta F_j(\mu) + g_k'(\epsilon').$$

Therefore

$$\frac{\partial}{\partial \mu} \log P(Y_i = 1|\mu) = \frac{-\frac{1}{\sigma} \sum_{j=1}^{k} \Delta f_j(\mu)}{\sum_{j=1}^{k} \Delta F_j(\mu)} + g'_k(\epsilon')$$

and

$$\frac{\partial}{\partial \mu} \log P(Y_i = 0|\mu) = \frac{\frac{1}{\sigma} \sum_{j=1}^{k} \Delta f_j(\mu)}{1 - \sum_{j=1}^{k} \Delta F_j(\mu)} + g'_k(\epsilon').$$

Substituting these into $I_{\mu,i}$, we obtain:

$$I_{\mu,i} = \frac{\left(\frac{1}{\sigma} \sum_{j=1}^{k} \Delta f_j(\mu)\right)^2}{\left(\sum_{j=1}^{k} \Delta F_j(\mu)\right)\left(1 - \sum_{j=1}^{k} \Delta F_j(\mu)\right)} + h_k(\epsilon'). \tag{71}$$

We now make use of the following result, which is a corollary of (Kipnis & Duchi, 2022, Lemma 11):

**Lemma H.3.** *Assume the hypotheses of Theorem 4.3. For every $\delta > 0$,*

$$\frac{\left(\frac{1}{\sigma} \sum_{j=1}^{k} \Delta f_j(\mu)\right)^{2+\delta}}{\left(\sum_{j=1}^{k} \Delta F_j(\mu)\right)^{1+\delta}\left(1 - \sum_{j=1}^{k} \Delta F_j(\mu)\right)^{1+\delta}} \leq \Gamma_\delta(\mu).$$

Using this in equation 71 completes the proof of Lemma D.1 $\qquad\square$

## I   Proof of Lemmas H.1 and H.2.

The proofs of Lemmas H.1 and H.2 rely on the following lemma, which we prove first.

**Lemma I.1.** *For $i = 1, 2$, with $\theta_i \in \mathbb{R}$, $\epsilon_i > 0$, and $\alpha_i$ as defined in equation 8, the following bound holds:*

$$P\left[\left|\alpha_i - \frac{\theta_i - \mu}{\sigma}\right| \geq \epsilon_i \left(\frac{\theta_i - \mu}{\sigma}\right)\right] \leq 2\exp\left(\frac{-\epsilon_i^2 \tilde{C}_i n_i}{3}\right).$$

*Here, $\tilde{C}_i$, is a constant that depends only on $F_X$, $\theta_i$, $\sigma$, and $\mu$, but is independent of $n_1, n_2, n_3$.*

*Proof.* For $i = 1, 2$, we want to show that

$$P\left[\left|\alpha_i - \frac{\theta_i - \mu}{\sigma}\right| \geq \epsilon_i \left(\frac{\theta_i - \mu}{\sigma}\right)\right] \leq 2\exp\left(\frac{-\epsilon_i^2 C_i n_i}{3}\right).$$

We recall the definition of $\alpha_i$ from equation 8 and write

$$\alpha_i = F_X^{-1}(\mathbb{E}(F_i) + G_i),$$

where $G_i \triangleq F_i - \mathbb{E}(F_i)$ is a random variable.

Note that

$$F_X^{-1}(\mathbb{E}(F_i)) = F_X^{-1}\left[\mathbb{E}\left(\frac{1}{n_i} \sum_{j=n_{i-1}+1}^{n_i+n_{i-1}} Y_j\right)\right] = \left(\frac{\theta_i - \mu}{\sigma}\right),$$

where we have assumed $n_0 = 0$. Hence we can write

$$P\left[\left|\alpha_i - \frac{\theta_i - \mu}{\sigma}\right| \geq \epsilon_i \left(\frac{\theta_i - \mu}{\sigma}\right)\right] = P\left[\left|F_X^{-1}(\mathbb{E}F_i + G_i) - F_X^{-1}(\mathbb{E}F_i)\right| \geq \epsilon_i \left(\frac{\theta_i - \mu}{\sigma}\right)\right].$$

Using the Taylor series expansion

$$F_X^{-1}(\mathbb{E}(F_i) + G_i) = F_X^{-1}(\mathbb{E}(F_i)) + \frac{1}{f_X(F_X^{-1}(\mathbb{E}(F_i)))} \cdot G_i + \mathcal{O}(G_i^2). \tag{72}$$

Thus equation 72 becomes

$$P\left[\left|\alpha_i - \frac{\theta_i - \mu}{\sigma}\right| \geq \epsilon_i\left(\frac{\theta_i - \mu}{\sigma}\right)\right] = P\left[\left|\frac{G_i}{f_X(F_X^{-1}(\mathbb{E}F_i))} + \mathcal{O}(G_i^2)\right| \geq \epsilon_i\left(\frac{\theta_i - \mu}{\sigma}\right)\right]. \tag{73}$$

For sufficiently large $n_i$, $G_i$ is small, and the term $\mathcal{O}(G_i^2)$ is at most $\frac{G_i}{f_X(F_X^{-1}(\mathbb{E}[F_i]))}$ and hence equation 73 becomes

$$P\left[\left|\frac{G_i}{f_X(F_X^{-1}(\mathbb{E}F_i))} + \mathcal{O}(G_i^2)\right| \geq \epsilon_i\left(\frac{\theta_i - \mu}{\sigma}\right)\right] \leq P\left[|G_i| \geq \frac{\epsilon_i}{2}\left(\frac{\theta_i - \mu}{\sigma}\right) f_X(F_X^{-1}(\mathbb{E}F_i))\right]. \tag{74}$$

Using the Chernoff bound for Bernoulli random variables form (Har-Peled, 2011, Theorem 27.17)), for all $0 \leq \delta \leq 1$, we have:

$$P\left[|F_i - \mathbb{E}(F_i)| \geq \delta\mathbb{E}(F_i)\right] \leq 2\exp\left(\frac{-\delta^2 n_i \mathbb{E}[F_i]}{3}\right). \tag{75}$$

Therefore, applying this bound to equation 74, we obtain:

$$P\left[|F_i - \mathbb{E}(F_i)| \geq \frac{\epsilon_i}{2}\left(\frac{\theta_i - \mu}{\sigma}\right) f_X(F_X^{-1}(\mathbb{E}F_i))\right]$$

$$\leq 2\exp\left[-\frac{n_i}{3(\mathbb{E}F_i)}\left(\frac{\epsilon_i^2}{4}\right)\left(\frac{\theta_i - \mu}{\sigma}\right)^2 f_X^2(F_X^{-1}(\mathbb{E}F_i))\right] = 2\exp\left(\frac{-n_i \epsilon_i^2 \tilde{C}_i}{3}\right),$$

where $\tilde{C}_i = \frac{1}{4(\mathbb{E}F_i)}\left(\frac{\theta_i - \mu}{\sigma}\right)^2 f_X^2(F_X^{-1}(\mathbb{E}F_i)) = \frac{1}{4\left(F_X\left(\frac{\theta_i - \mu}{\sigma}\right)\right)}\left(\frac{\theta_i - \mu}{\sigma}\right)^2 f_X^2\left(\frac{\theta_i - \mu}{\sigma}\right)$.

Finally we can write:

$$P\left[\left|\alpha_i - \frac{\theta_i - \mu}{\sigma}\right| \geq \epsilon_i\left(\frac{\theta_i - \mu}{\sigma}\right)\right] \leq 2\exp\left(\frac{-n_i \epsilon_i^2 \tilde{C}_i}{3}\right).$$

This completes the proof. □

## I.1 Proof of Lemma H.1

From Lemma I.1, for $i = 1, 2$ and $\epsilon_i > 0$, the following inequality holds:

$$P\left[\left|\alpha_i - \frac{\theta_i - \mu}{\sigma}\right| \leq \epsilon_i\left(\frac{\theta_i - \mu}{\sigma}\right)\right] \leq 1 - 2\exp\left(\frac{-n_i \epsilon_i^2 \tilde{C}_i}{3}\right).$$

This implies that as $n_i \to \infty$, $\alpha_i$ becomes arbitrarily close to $\frac{\theta_i - \mu}{\sigma}$ within a margin proportional to $\epsilon_i$.

For some $\epsilon_1 > 0$, if we have

$$\left|\alpha_i - \frac{\theta_i - \mu}{\sigma}\right| \leq \epsilon_1\left(\frac{\theta_i - \mu}{\sigma}\right), \tag{76}$$

for $i = 1, 2$, then

$$\alpha_1 - \alpha_2 \in \left(\left(\frac{\theta_1 - \mu}{\sigma}\right) - \left(\frac{\theta_2 - \mu}{\sigma}\right)\right)(1 \pm 2\epsilon_1),$$

and

$$\frac{1}{\left(\left(\frac{\theta_1-\mu}{\sigma}\right)-\left(\frac{\theta_2-\mu}{\sigma}\right)\right)}\frac{1}{(1+2\epsilon_1)} \leqslant \frac{1}{(\alpha_1-\alpha_2)} \leqslant \frac{1}{\left(\left(\frac{\theta_1-\mu}{\sigma}\right)-\left(\frac{\theta_2-\mu}{\sigma}\right)\right)}\frac{1}{(1-2\epsilon_1)}, \tag{77}$$

or equivalently,

$$\frac{\left(\left(\frac{\theta_1-\mu}{\sigma}\right)\theta_2-\left(\frac{\theta_2-\mu}{\sigma}\right)\theta_1\right)(1-2\epsilon_1)}{\left(\left(\frac{\theta_1-\mu}{\sigma}\right)-\left(\frac{\theta_2-\mu}{\sigma}\right)\right)(1+2\epsilon_1)}-\mu \leqslant \left(\frac{\alpha_1\theta_2-\alpha_2\theta_1}{\alpha_1-\alpha_2}\right)-\mu \leqslant \frac{\left(\left(\frac{\theta_1-\mu}{\sigma}\right)\theta_2-\left(\frac{\theta_2-\mu}{\sigma}\right)\theta_1\right)(1+2\epsilon_1)}{\left(\left(\frac{\theta_1-\mu}{\sigma}\right)-\left(\frac{\theta_2-\mu}{\sigma}\right)\right)(1-2\epsilon_1)}-\mu. \tag{78}$$

Simplifying, we conclude that if equation 76 holds, then

$$\frac{-4\epsilon_1}{(1+2\epsilon_1)}\cdot\mu \leqslant \hat{\mu}_c-\mu \leqslant \frac{4\epsilon_1}{(1-2\epsilon_1)}\cdot\mu. \tag{79}$$

This implies that for all sufficiently small $\epsilon_1$,

$$P\left[|\mu-\hat{\mu}_c| \geqslant |\mu|\frac{4\epsilon_1}{(1-2\epsilon_1)})\right] \leqslant P\left[\left|\alpha_1-\frac{\theta_1-\mu}{\sigma}\right| \geqslant \epsilon_1\left(\frac{\theta_1-\mu}{\sigma}\right)\right]+P\left[\left|\alpha_2-\frac{\theta_2-\mu}{\sigma}\right| \geqslant \epsilon_1\left(\frac{\theta_2-\mu}{\sigma}\right)\right].$$

Then, applying Lemma I.1, we can write:

$$\begin{aligned}P\left[|\mu-\hat{\mu}_c| \geqslant |\mu|\frac{4\epsilon_1}{(1-2\epsilon_1)})\right] &\leqslant 2\exp\left(\frac{-n_1\epsilon_1^2\tilde{C}_1}{3}\right)+2\exp\left(\frac{-n_2\epsilon_1^2\tilde{C}_2}{3}\right)\\ &= 2\exp\left(\frac{-n_1\epsilon_1^2\tilde{C}_1}{3}\right)\left[1+\exp\left(\frac{-n_1\epsilon_1^2\eta}{3}\right)\right].\end{aligned} \tag{80}$$

Let $n_1 = n_2 = \frac{n_3}{\log n_3}$ and $\epsilon_1 = \frac{\log n_3}{\sqrt{n_3}}$. Then, we have:

$$P\left[|\mu-\hat{\mu}_c| \geqslant |\mu|\left(\frac{4\log n_3}{\sqrt{n_3}-2\log n_3}\right)\right] \leqslant 2\exp\left(\frac{-\tilde{C}_1\log n_3}{3}\right)\left[1+\exp\left(\frac{-\eta\log n_3}{3}\right)\right]. \tag{81}$$

Now from equation 81, we conclude that $\mu$ converges in probability to $\mu$ as $n_3 \to \infty$. This completes the proof.

$\square$

## I.2 Proof of Lemma H.2

Now we recall the definition of $\hat{\sigma}_c$ and write

$$(\hat{\sigma}_c-\sigma) = \sigma\left(\frac{\theta_1-\theta_2}{\sigma(\alpha_1-\alpha_2)}-1\right). \tag{82}$$

The approach is very similar to that of the Lemma H.1 and we omit minor calculations. If we have

$$\left|\alpha_i-\frac{\theta_i-\mu}{\sigma}\right| \leqslant \epsilon_1\left(\frac{\theta_i-\mu}{\sigma}\right),$$

for $\epsilon_1 > 0$ and $i = 1,2$, then

$$\frac{-2\epsilon_1\sigma}{(1+2\epsilon_1)} \leqslant \hat{\sigma}_c-\sigma \leqslant \frac{2\epsilon_1\sigma}{(1-2\epsilon_1)}.$$

We can apply the same union bound approach used in proving Lemma H.1, yielding:

$$P\left[|\sigma - \hat{\sigma}_c| \geq |\sigma|\frac{2\epsilon_1}{(1-2\epsilon_1)}\right] \leq P\left[\left|\alpha_1 - \frac{\theta_1 - \mu}{\sigma}\right| \geq \epsilon_1\left(\frac{\theta_1 - \mu}{\sigma}\right)\right] + P\left[\left|\alpha_2 - \frac{\theta_2 - \mu}{\sigma}\right| \geq \epsilon_1\left(\frac{\theta_2 - \mu}{\sigma}\right)\right].$$

Using the value of $\epsilon_1 = \frac{\log n_3}{\sqrt{n_3}}$ and following a similar approach as in Lemma H.1, we apply Lemma I.1 to obtain the following:

$$P\left[|\sigma - \hat{\sigma}_c| \geq |\sigma|\frac{2\log n_3}{\sqrt{n_3} - 2\log n_3}\right] \leq 2\exp\left(\frac{-\tilde{C}_1 \log n_3}{3}\right)\left[1 + \exp\left(\frac{-\eta \log n_3}{3}\right)\right].$$

Where $\tilde{C}_1$ and $\eta$ are the constants as defined in Lemma H.1. Hence, as $n \to \infty$, $\hat{\sigma}_c$ converges in probability to $\sigma$. This concludes the proof of the Lemma. $\qquad\square$

*Remark* I.2. In the proofs of Lemmas H.1 and H.2 we only require that $\hat{\mu}_c$ and $\hat{\sigma}_c$ converge in probability to $\mu$ and $\sigma$, respectively. This is ensured as soon as the product $n_1\epsilon_1^2$ tends to infinity, since the deviation probabilities then go to zero for every fixed threshold. In particular, our choice

$$n_1 = n_2 = \frac{n_3}{\log n_3}, \qquad \epsilon_1 = \frac{\log n_3}{\sqrt{n_3}},$$

yields $n_1\epsilon_1^2 = \log n_3 \to \infty$ and is therefore sufficient for the consistency in probability established in Lemmas H.1 and H.2. One may also consider the more general allocation

$$n_1 = n_2 = \frac{n_3}{(\log n_3)^{1-\delta}}, \qquad 0 < \delta < 1,$$

with the same choice $\epsilon_1 = \frac{\log n_3}{\sqrt{n_3}}$. In this case

$$n_1\epsilon_1^2 = \frac{n_3}{(\log n_3)^{1-\delta}} \cdot \frac{(\log n_3)^2}{n_3} = (\log n_3)^{1+\delta},$$

so the deviation probabilities $P(A_{n_3})$ satisfy

$$P(A_{n_3}) \leq \exp\left(-C(\log n_3)^{1+\delta}\right), \qquad C > 0.$$

The series

$$\sum_{n_3=1}^{\infty} \exp\left(-C(\log n_3)^{1+\delta}\right)$$

is then finite, and the Borel–Cantelli lemma (see, e.g., (Durrett, 2019, Theorem 2.3.7)) implies that the events $A_{n_3}$ occur only finitely many times almost surely. Under this alternative scaling, the deviation probabilities decay fast enough to yield a stronger conclusion for Lemmas H.1 and H.2, namely almost sure convergence instead of convergence in probability. Since none of our results require this stronger mode of convergence, we work with the simpler choice $n_1 = n_2 = n_3/\log n_3$, which already ensures the consistency guarantees established in Lemmas H.1 and H.2.

