# OpenReview forum: "One-Bit Distributed Mean Estimation with Unknown Variance"
_TMLR — Accepted by TMLR_

### Review · Reviewer_LzHk · 2025-11-20

**Summary Of Contributions:**

The paper provides several algorithms to solve estimation of the mean in location scale families in distributed context. The key constraint is that the distributed observers of samples can send only one bit to the central server.

The algorithms naturally split into the simpler non-adaptive and more complex adaptive classes. It turns out that the proposed algorithms send the bits just based on a simple comparisons with a threshold.

The problems considered are non-trivial. The authors achieved substantial progress in advancing state-of-the-art. Asymptotic results included are proved using advanced statistical techniques. The review of the state-of-the-art is up-to-date. The quality of presentation is excellent.

**Audience:**

Yes

**Audience Explanation:**

Estimation of the mean in various contexts is a foundational problem of estimation theory. One can find an example of a paper with similar topic in this year's TMLR:

Vardhan, Mazumdar: Collaborative Compressors in Distributed Mean Estimation with Limited Communication Budget
https://openreview.net/pdf?id=AtCKHCoMA7

**Broader Impact Concerns:**

I can see no ethical implications of this work.

**Claims And Evidence:**

Yes

**Claims Explanation:**

The new results in the paper are of several kinds.

- new algorithms,
- study of  their asymptotic behavior
- simulations of algorithms in finite samples.

Since the authors opted to put proofs into the appendixes, it seems that the simulations are the key part of the paper. The authors provided the code to reproduce their experiments in the supporting material.

**Requested Changes:**

CRITICAL:

1. I do not understand the series of inequalities below equation (60) on page 31. If $F_X$ is indeed $L$-Lipschitz, then there should be no $\sigma$ in the middle term. For the last inequality to hold, one presumably assumes that $z$ belongs to the interval $[\theta_{j-1}, \theta_j]$ or something similar.

TO STRENGTHEN WORK:

2. I would like to ask the authors, whether there could be applications beyond distributed computing, e.g. for price estimation in economics, finance, or even sociology, since these fields often uses location-scale families. If so, then the authors could sketch further interdisciplinary applications.

3. References to the algorithms could be more polished. As an example on page 10 the authors refer to the non-adaptive protocol of Section 3. However, there are 2 non-adaptive protocols in Section 3 - see pages 6 and 9.

4. It may be a good idea to include location-scale family as part of the paper's title.

5. I would myself choose another name for $Y_i$. For me, $Y_i$ would a real-valued random variable, whereas the underlying random variable is Boolean. Maybe $B_i$ or $M_i$ would be a (admittedly marginally) better choice, if one insists on random variables being capitalized? This is however more a suggestion for future work, as large-scale paper changes like this tend to lead to omissions.

---

> ### Author Response · Authors · 2025-11-24
> **Response to review LzHk**
>
> We thank the reviewer for the detailed and encouraging feedback, and sharing the reference paper of Vardhan and Mazumdar. We were not aware of this paper, and it seems to have been published only around the time of our submission. Their setup is slightly different: This seems to be more in line with what we discuss in the last paragraph of Section 2.3. We will nevertheless go through this reference carefully, and cite it in the final version.
>
> ### Response to individual comments:
> 1. Regarding the critical comment: We thank the reviewer for pointing this out. As you rightly noticed, the statement holds only when $z\in[\theta_{j-1},\theta_j]$ (which is what we require), and we apologise for any confusion. We will clarify this in the final version.
> 2. Regarding applications beyond distributed computing: We thank the reviewer for the suggestion. We have not thought about applications beyond distributed computing, but will try to look for connections to these fields and include any relevant references. While scale-location families are widely used in many areas as you rightly pointed out, we are not aware of any problems in these areas that require one-bit communication constraints or even a distributed setup. We will however explore this further and perhaps add a remark in the final version.
> 3. Regarding the references: We apologise for the confusion, and will make the references more precise in the final version.
> 4. Regarding the change in title: We thank the reviewer for the suggestion. To be concise, we would prefer to keep the current title, but will definitely amend the abstract and introduction to explicitly state that the results apply to location–scale
> families
> 5. Regarding the notation: To remain consistent with the pseudocode and earlier sections, we would prefer to retain the notation. However, we will reconsider this in future work.

---

### Review · Reviewer_6jTy · 2025-11-20

**Summary Of Contributions:**

This is a very technical paper, studying the problem of distributed mean estimation in the case where clients communicate only a bit to the server doing the estimation, derived from a pre-decided threshold. The authors specifically study the case in which both mean and variance are unknown. The authors study both the non-adaptive case (a single fixed threshold,no coordination), and the adaptive case (for them, 2 thresholds and 2 rounds, with the 2 thresholds participating in only one round each respectively), and show that the lower bound for the non-adaptive case is strictly higher than the adaptive case for a range of symmetric, log-concave distributions, including generalized gaussians with  $1 < β < 1.85$. The authors are able to quantify the gap, which neat.

As a byproduct the authors also derive a non-adaptive protocol for the standard deviation and derived its asymptotic MSE, but explicitly do not claim optimality.

**Audience:**

Yes

**Audience Explanation:**

I think there's a technical subset of the TMLR audience which can either directly use the technique or can use the proof techniques demonstrated.

**Claims And Evidence:**

Yes

**Claims Explanation:**

I went throught he proofs and while I am not an expert in this type of analysis, I think I could follow them and did not find any major mistakes, except one nit (unless I made a mistake):

In appendix I.1, I think you need to have $\frac{\tilde{C}_1}{3}​​>1$ to hold in order to claim strong consistency/almost sure convergence, but I don't think there's an assumption placed on $\tilde{C}$ anywhere? It doesn't change the main point of the paper, but I think either you need to e.g. choose $n_1=\frac{n_3}{(\log n_3) ^{1-\delta}}$ with $\delta >0$  to ensure that it remains controlled, or change the claim to weak convergence.

**Requested Changes:**

Non, except maybe the nit.

---

> ### Author Response · Authors · 2025-11-24
> **Response to reviewer 6jTy**
>
> We sincerely thank the reviewer for carefully going through the paper and giving feedback.
>
> ### Regarding the comment on $\tilde{C}_i$ in Appendix I:
> We would like to clarify that we are not claiming strong convergence (almost sure convergence, in the terminology of the paper) of $\alpha_i$ to $\frac{\theta_i-\mu}{\sigma}$ as $n_i\to\infty$. We only require convergence in probability, and this holds as long as $\tilde{C}_i>0$. This statement is stronger than weak convergence (or convergence in distribution), and this suffices to prove Lemmas H.1 and H.2, which in turn is required by Lemma C.1 where we need to show that $\sqrt{n_3}|\sigma-\hat{\sigma}_c||\mu-\hat{\mu}_c|$ converges to zero *in probability* (we do not require strong convergence). This suffices to derive our bound on the MSE of $\hat{\mu}_f$.
> We hope that this clarifies your point.
>
> We however agree with your point that changing $n_1=n_2=\frac{n_3}{(\log n_3)^{1-\delta}}$ would additionally give us strong convergence of $\alpha_i$. We will add a remark about this in the final version of our paper.

---

> > ### Comment · Reviewer_6jTy · 2025-12-09
> >
> > thank your for your response and correction of my mixup

---

### Review · Reviewer_iCZe · 2025-11-30

**Summary Of Contributions:**

This paper considers the problem of distributed mean estimation, whereby a central server is tasked with estimating the mean of samples when each user has access to one i.i.d. sample and is restricted to sending only a single bit. The paper distinguishes between two types of protocols (adaptive and non-adaptive) and analyzes the mean estimation problem by bounding the mean squared error as the number of users goes to infinity.

**Strengths**

* The paper is very well written. The problem is clearly presented, and the authors make their contributions explicit.
* The problem is interesting and potentially useful in resource-constrained environments. The analysis is thorough and provides meaningful insight into the quality of the proposed protocols.
* In addition to theory, the numerical results illustrate how the estimators behave in practice.

**Weaknesses**

* It would be beneficial to comment on how the problem changes if users are allowed to send 2 bits per sample. Although a variant of this is briefly mentioned in the future-work section, a more explicit discussion of the marginal or qualitative benefits (if any) of the 2-bit case would help justify the focus on the 1-bit regime.
* How do the analytical asymptotic results inform performance in the finite-user setting? Based on the numerical experiments, the protocols appear to achieve relatively small MSE even for moderate numbers of users. A discussion connecting the asymptotic theory to finite-sample behavior is currently missing.
* There should also be some discussion of the communication or computation cost associated with the adaptive protocol.

**Audience:**

Yes

**Audience Explanation:**

The problem is related to similar problems in federated learning, privacy-preserving statistics, and sensor networks, and would be of interest to those interested in these topics.

**Broader Impact Concerns:**

No direct ethical or societal risks are anticipated.

**Claims And Evidence:**

Yes

**Claims Explanation:**

To my reading, all the theorems are based on solid mathematical proofs.

**Requested Changes:**

Please address the weaknesses that I highlighted above. I believe that the paper presents an interesting problem and develops nice analysis. I would recommend acceptance if the authors can address the weaknesses.

---

> ### Author Response · Authors · 2025-12-05
> **Response to reviewer iCZe**
>
> We sincerely thank the reviewer for the positive assessment of the paper and for the constructive suggestions.
>
> ### Increasing the number of bits per user
> We appreciate this suggestion. While some of the techniques we use in this paper are specific to the $1$-bit case (the protocols designed, and some tools used for the lower bounds), it should be possible to use similar approaches to derive upper and lower bounds on the MSE with $2$-bit and general $B$-bit protocols for $B\geq 2$. However, for a broad class of symmetric log-concave distributions, increasing $B$ will only marginally improve the mean squared error, as we outline below.
>
> Using the Van Trees inequality, we can derive a Bayesian lower bound on the asymptotic MSE achieved *without* any communication constraints (i.e., $B=\infty$) in terms of the Fisher information. For any $B>1$, the optimal MSE for a $B$-bit communication protocol would be sandwiched between this bound and that of the corresponding bounds for $1$-bit protocols. We have computed these numerically, and they are found to be very close. To give you a concrete example, for the gaussian scale-location family, the asymptotic MSE achieved by our adaptive protocol is $\approx 1.1\sigma^2/n$, whereas the corresponding Bayesian lower bound even without any communication constraints is $\approx 0.91\sigma^2/n$, which indicates that allocating more bits can only give marginal improvements.
>
> We thank the reviewer for pointing us in this direction, and will surely emphasize this in the final version of our paper. We will supplement our results (simulation plots and discussions) with the above lower bound and hope that this gives sufficient justification and motivation for studying $1$-bit protocols.
>
> ### How do the analytical asymptotic results inform performance in the finite-user setting
> Our techniques allow us to derive the first-order term in the MSE
> $$\mathrm{MSE}(n)=\frac{C}{n}+o(1/n),$$
> where $C$ is exactly the constant derived in our analysis. Deriving the higher order terms (which could be significant for very small values of $n$) could be more challenging, and would require other techniques. However, as seen from our simulation results (See Figures 5--8), these bounds are reasonably good even for moderately large $n$, and hence our analysis gives good predictions even in this regime. We will include a short discussion on this in the final manuscript.
>
> ### Communication and computational complexity of our protocols
> Our protocols are extremely lightweight: In both the adaptive and non-adaptive protocols, each user transmits exactly one bit, and hence the total communication complexity is only $O(n)$, where $n$ is the total number of users.
>
> For the adaptive protocol, if we assume that each user $i$ is able to listen to the transmission of the first $i-1$ users, then no further communication by the server is required (since the last $n_3$ users can directly compute $\hat{\mu}_c$ by observing the first $n_1+n_2$ bits); If they cannot listen to the first few transmissions, then the server needs to broadcast only a single real number to the remaining $n_3$ users and hence requires minimal communication. Please see the footnotes in Page 10 regarding this.
>
> Likewise, the computational complexity is also minimal: In both protocols, each user does simple thresholding which requires $O(1)$ computational cost. At the server, the overall complexity is only $O(n)$: this requires reading and averaging $O(n)$ bits, and a constant number of real-valued operations.
>
> We will add a brief discussion clarifying this in the final version.

---

> > ### Comment · Reviewer_iCZe · 2025-12-09
> > **response has addressed all my questions**
> >
> > I would like to thank the authors for their detailed response to my questions. The responses have clarified my questions/suggestions, and I maintain my positive rating of the paper. I would encourage the authors to include some of the discussion in the response, as suitable, to the main paper (e.g., I have also noted that Figure 5-8 are quite good for moderate n, which indicates reasonable performance in finite-user setting).

---

> > > ### Author Response · Authors · 2025-12-10
> > > **Response**
> > >
> > > Thank you. We will surely include the above in the final version of the paper.

---

### Author Response · Authors · 2025-12-17
**Revised manuscript**

We have uploaded a revised version of our manuscript.

Here is a list of major changes in the revised version:
- Page 2, two paragraphs added prior to Section 1.1 with potential applications beyond distributed learning, computing and optimization (in response to reviewer LzHk)
- Page 10, last paragraph added clarifying subsequent references to 'non adaptive protocol' (in response to reviewer LzHk)
- Page 14, Section 4.4 added clarifying the communication and computational complexity of our protocols (in response to reviewer iCZe)
- Page 14, Section 4.5 added comparing our bounds with the centralized (no communication constraints) case, and demonstrating that increasing the number of bits only marginally improves the MSE (in response to reviewer iCZe)
- Page 18, Remark 5.1 added to comment on the finite length performance of protocols, and that the MSE is almost $O(1/n)$ for even moderately large $n$ (response to reviewer iCZe)
- Page 45, Remark I.2 added to indicate that with a slightly different choice of $n_1,n_2$, it is possible to achieve almost sure convergence of $\alpha_i$, but this is not essential to our proofs (in response to reviewer 6jTy)
- Page 35, a little after (60), we have clarified that $z\in[\theta_j,\theta_{j+1}]$

Additionally, we have made minor changes/corrections to the rest of the paper.

We hope that the revised version has addressed all the comments of the reviewers. We would like to once again thank them for their insightful feedback that has helped improve our paper.

---

### Decision · Action_Editor_tzon · 2026-01-16

**Recommendation:** Accept as is

**Additional Comments:**

The paper proposes the non-adaptive and adaptive protocols for distributed mean estimation problem with 1-bit communication constraints with unknown variance, and provides solid theoretical analysis of these protocols. The paper also provides a lower error bound that justifies the benefit of the proposed adaptive protocol. The analysis is comprehensive.

**Audience:**

Yes

**Audience Explanation:**

The paper studies the distributed mean estimation problem with 1-bit communication constraints with unknown variance. Distributed mean estimation is a fundamental problem in machine learning and of strong interest to the TMLR audience.

**Claims And Evidence:**

Yes

**Claims Explanation:**

The proofs of the theoretical results are provided, and the results are further validated through numerical experiments.